# Cortico-cerebellar networks
# as decoupling neural interfaces

**Joseph Pemberton**\*
Bristol Computational Neuroscience Unit
Dept. of Computer Science, SCEEM
University of Bristol, UK
`oq19042@bristol.ac.uk`

**Ellen Boven**\*
Bristol Computational Neuroscience Unit
School of Phys., Pharm. and Neuroscience
University of Bristol, UK
`ellen.boven@bristol.ac.uk`

**Richard Apps**
School of Phys., Pharm. and Neuroscience
University of Bristol, UK
`r.apps@bristol.ac.uk`

**Rui Ponte Costa**
Bristol Computational Neuroscience Unit
Dept. of Computer Science, SCEEM
University of Bristol, UK
`rui.costa@bristol.ac.uk`

## Abstract

The brain solves the credit assignment problem remarkably well. For credit to be assigned across neural networks they must, in principle, wait for specific neural computations to finish. How the brain deals with this inherent locking problem has remained unclear. Deep learning methods suffer from similar locking constraints both on the forward and feedback phase. Recently, decoupled neural interfaces (DNIs) were introduced as a solution to the forward and feedback locking problems in deep networks. Here we propose that a specialised brain region, the cerebellum, helps the cerebral cortex solve similar locking problems akin to DNIs. To demonstrate the potential of this framework we introduce a systems-level model in which a recurrent cortical network receives online temporal feedback predictions from a cerebellar module. We test this cortico-cerebellar recurrent neural network (ccRNN) model on a number of sensorimotor (line and digit drawing) and cognitive tasks (pattern recognition and caption generation) that have been shown to be cerebellar-dependent. In all tasks, we observe that ccRNNs facilitates learning while reducing ataxia-like behaviours, consistent with classical experimental observations. Moreover, our model also explains recent behavioural and neuronal observations while making several testable predictions across multiple levels. Overall, our work offers a novel perspective on the cerebellum as a brain-wide decoupling machine for efficient credit assignment and opens a new avenue between deep learning and neuroscience.

## 1 Introduction

Efficient credit assignment in the brain is a critical part of learning. However, how the brain solves the credit assignment problem remains a mystery. One of the central issues of credit assignment across multiple stages of processing is the need to wait for later stages to finish their computation before learning can take place. In deep artificial neural networks, where processing is divided into a forward (i.e. prediction) and feedback (i.e. error backpropagation) phase, these constraints are explicit [1–5]. Given hidden activity $h$ at a certain layer or timestep, network computations

---

\*Equal contributions

35th Conference on Neural Information Processing Systems (NeurIPS 2021).

at subsequent layers/timesteps must first be computed before feedback gradients $\delta h$ are available for network parameters to be updated. Not only does this introduce computational inefficiency, in particular for the temporal case where backpropagation through several timesteps is required, but it is also considered biologically unrealistic [6]. Recently, a framework was introduced to decouple forward and feedback processing in artificial neural networks – decoupled neural interfaces (DNI; Jaderberg et al. [5])[2] – in which a connected *but distinct* neural network provides the main network with predicted forward activity or feedback gradients, thereby alleviating the locking problems.

Here, we propose that a specialised brain area, the cerebellum, performs a similar role in the brain. In the classical view the cerebellum is key for fine motor control and learning by constructing internal models of behaviour [7–11]. More recently, however, the idea that the cerebellum is also involved in cognition has gained significant traction [12–14]. An increasing body of behavioural, anatomical and imaging studies points to a role of the cerebellum in cognition in both human and non-human primates [12, 14–16]. In particular cerebellar impairments have been observed in language [15], working memory [17], planning [18], and others modalities [19]. Coupled with its notoriously uniform structure, these observations suggests that the cerebellum implements a *universal function* across the brain [7–9, 20]. Moreover, experimental studies looking at cortico-cerebellar interactions have directly demonstrated that cerebellar output is crucial for the development and maintenance of neocortical states [17, 21, 22]. However, to the best of our knowledge, no computational formulation exists which explicitly describes the function of such interactions between the cerebellum and cortical areas.

With this in mind and inspired by deep learning DNIs we introduce a systems-level model of cortico-cerebellar loops. In this model and consistent with the cerebellar universal role, the cerebellum serves to break the spatio-temporal locks inherent to both feedforward and feedback information processing in the brain, akin to DNI. In particular, we posit that the cerebellar forward internal model hypothesis of the cerebellum is equivalent to DNI-mediated unlocking of feedback gradients. Following this view the cerebellum not only provides motor or sensory feedback estimates, but also any other modality encoded by a particular brain region. In this regard we introduce a cortico-cerebellar RNN (ccRNN) model which we test on sensorimotor tasks: (i) a simple line drawing task [23–25] and (ii) drawing tasks with temporally complex input based on the MNIST dataset, but also (iii) on a cognitive task – caption generation [15]. Our results support the decoupling view of cortico-cerebellar networks by showing that they improve learning while reducing ataxia-like behaviours in a range of tasks, being qualitatively consistent with a wide range of experimental observations [15, 23–25].

## 2   Cerebellum as a cortical feedback prediction machine

We first describe DNIs following Jaderberg et al. [5] and then establish the link to cortico-cerebellar networks. We focus on "backward" (feedback) DNI which directly facilitates learning and is the model considered in this paper, but we also consider "forward" DNI to be strongly relevant to cerebellar computation (see SM A).

Although in this paper we focus on the DNI in a temporal setting, to better explain DNI we first briefly explain the spatial case. This case not only serves as a general basis for the temporal case (i.e. one can think of a recurrent neural network as a specific instance of a feedforward network), but also highlights the possibility to place cerebellar modules between different pathways. Assume that a feedforward neural network consists of $N$ layers, with the $i$th layer ($1 \leq i \leq N$) performing a "computational step" $f_i$ with parameters $\theta_i$. Given input $x$ at layer 1, the output of the network at its final layer is therefore given by $f_N(f_{N-1}(\ldots f_2(f_1(x))\ldots))$. Let $\mathcal{F}_i^j$ denote the composition of steps from layer $i$ to layer $j$ (inclusively). Finally, let $h_i$ denote the (hidden) activity at layer $i$, so that $h_i = f_i(h_{i-1})$ with $h_0 = x$.

We now illustrate the learning constraints of standard artificial neural networks used in deep learning. Suppose that a network is in its feedback (or backpropagation) phase, with current input-target pair $(x, y)$. To update the layer parameters $\theta_i$ the gradient $\frac{dL}{d\theta_i}$ is required, where $L = \mathcal{L}(y, \hat{y})$ is the loss which compares the target value against the model output $\hat{y} = \mathcal{F}_1^N(x)$ under some loss function $\mathcal{L}$; we then apply gradient descent on the parameters, $\theta_i \leftarrow \theta_i - \alpha \frac{dL}{d\theta_i}$, with learning rate $\alpha > 0$. Suppose however that the network has only recently received the input $x$ and is currently only at layer $i$ of

---

[2]DNIs are related to earlier work on using network critics to train neural networks [2].

the forward computation. In order to update the corresponding parameters of that layer, $\theta_i$, the layer must first wait for all remaining layers to finish $f_j$ $(j > i)$ for the loss to be computed. Only then are the various gradients of the loss are backpropagated and $\frac{dL}{d\theta_i}$ is finally available. The enforced wait makes layer $i$ "feedback locked"[3] to $\mathcal{F}_{i+1}^N$ and imposes an complete dependence between learning of the layer in question and the computations in the remaining network.

The basis of DNI is to break the feedback lock by sending a copy of the hidden layer $i$ activity $h_i$ to a separate neural network $\mathcal{C}$, termed "synthesiser" in [5] and which we propose as being implemented by the cerebellum in the brain, which then returns a *synthetic gradient* $\hat{\delta}_i$ – an estimate of the real loss gradient with respect to the layer, i.e. $\mathcal{C}_i(h_i) = \hat{\delta}_i \approx \delta_i$ where $\delta_i := \frac{dL}{dh_i}$. We can then update our original layer immediately according to gradient descent, effectively breaking the feedback lock with

$$\theta_i \leftarrow \theta_i - \alpha_{\mathcal{C}} \hat{\delta}_i \frac{dh_i}{d\theta_i}; \qquad \hat{\delta}_i = \mathcal{C}_i(h_i) \tag{1}$$

How can we trust the synthetic gradient to improve performance? The parameters of $\mathcal{C}$ are themselves learnt so as to best approximate the observed feedback gradient. That is, once the standard forward and feedback computations are performed in the rest of the network (which may take some time) and $\frac{dL}{dh_i}$ is available, we minimise the objective function $\mathcal{L}_{\mathcal{C}} = ||\hat{\delta}_i - \frac{dL}{dh_i}||$. Assuming $\mathcal{C}$ learns a decent approximation, the original layer parameters should then move (roughly) along the direction of the true loss gradient. Consistent with the need for this approximation a model with a fixed cerebellum $\mathcal{C}$ (i.e. not learnt) gives poor or often no learning, indicating that $\mathcal{C}$ output is not simply facilitating learning via a stabilising effect (Fig. S7).

**Temporal feedback in RNNs**    Using the same terminology but replacing layer indices $i$ with time $t$ and setting $\theta_t = \theta \, \forall t$, we can apply the above to recurrent neural networks (RNNs; Fig. 1a). Now the synthesiser $\mathcal{C}$ predicts the effect of the current hidden activity $h_t$ to future losses $L_{\tau > t} = \sum_{\tau > t} L_\tau$, effectively providing the RNN with (approximate) temporal feedback before the sequence may even be finished. As an optimization algorithm to obtain the *true* temporal feedback $\delta_t$ we use backpropagation through time (BPTT). In principle, this feedback is only available once all the future feedback is available and $\mathcal{C}$ itself would be feedback locked. To circumvent this issue $\mathcal{C}(h_i)$ learns using a mixture of nearby true feedback signals and a bootstrapped (self-estimation) term (Fig. 1b), where for each $t$ we now minimise $||\hat{\delta}_t - \bar{\delta}_t||$ with $\bar{\delta}_t = \sum_{\tau > t}^T \frac{dL_\tau}{dh_t} + \mathcal{C}(h_T)\frac{dh_T}{dh_t}$ [5]. Here the first term includes the nearby feedback signals backpropagated within some limited time horizon $T$, and the second term includes the bootstrapped cerebellar prediction which estimates gradient information beyond this horizon (analogous to temporal difference algorithms used to estimate value functions in reinforcement learning). In this case the cerebellum feedback facilitates learning by enabling the RNN to learn using (predicted) future feedback signals that would otherwise be inaccessible, enabling a future-aware online learning setup. In particular, this model has already been demonstrated to thrive in the harsh but more biologically reasonable setting of reduced temporal windows for BPTT (small $T$; cf [5]).

## 2.1    Mapping between cortico-cerebellar networks and decoupled neural interfaces

Building on the temporal feedback DNI we introduce a systems model for cortico-cerebellar computation, which we denote as ccRNN (cortico-cerebellar recurrent neural network; Fig. 1). The model includes two principal components: an RNN which models a recurrent cortical network (or area, e.g. motor cortex or prefrontal cortex), and a feedforward neural network - the synthesiser $\mathcal{C}$ - which receives RNN activity before returning predicted feedback $\hat{\delta}_t$ and which we interpret as the cerebellum. The cerebellum receives the current cortical activity $h_t$ through its mossy fibres which then projects onto granule cells and Purkinje cells and sends back the predicted cortical feedback $\hat{\delta}_t$.

At the architectural level our model is in keeping with the observed anatomy of the cerebral-cerebellar circuitry. Recurrent connections are widespread in the neocortex and are considered functionally important for temporal tasks [26], whereas the cerebellum exhibits a relatively simple feedforward architecture [7–9]. Moreover, an important defining feature of the cerebellum is its expansion at the granular layer: with $> 50$ billion granule cells (more than the rest of the brain's neurons combined)

---

[3]We use feedback locked networks to incorporate both the "update" and "backward" lock as described in [5].

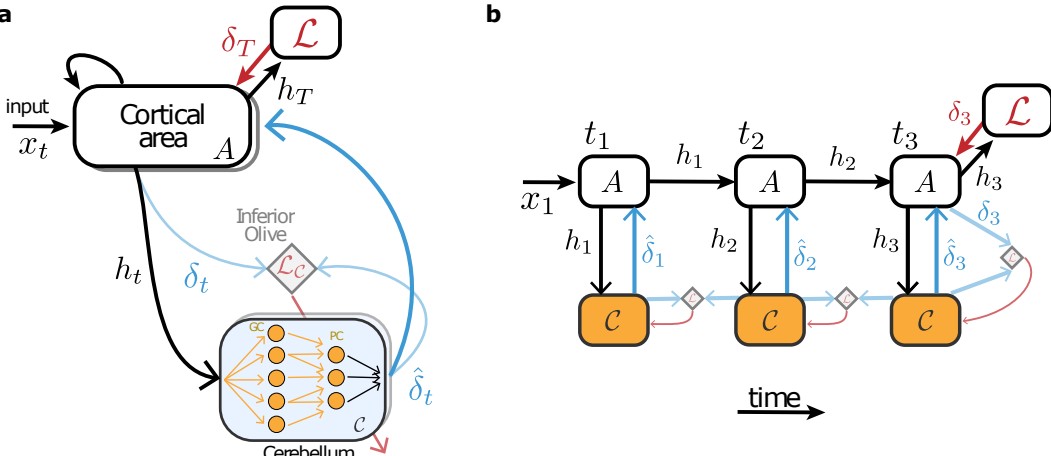

Figure 1: Cerebellum as a cortical feedback prediction machine. (**a**) Learning in a given recurrent cortical area $A$ is modelled as feedback-encoding gradients (red arrow) originating from a loss or error region $\mathcal{L}$ at the end of a task (time $T$). Meanwhile the cerebellum receives the current cortical activity $h_t$ (black arrow) which projects onto granule cells (GC; orange) and Purkinje cells (PC; orange) and sends back the predicted cortical feedback $\hat{\delta}_t$ (blue arrow). Cerebellar learning is mediated by the inferior olive, which compares estimated feedback $\hat{\delta}_t$ with observed feedback $\delta_t$, computing $\mathcal{L}_\mathcal{C} \sim ||\delta_t - \hat{\delta}_t||$ (see text for more details). (**b**) Example of cortico-cerebellar model unfolded across three time steps. Starting at $t_3$ the cortical network $A$ receives feedback $\delta_3$ (red), which is then transmitted to the cerebellar network as cortical feedback $\delta_3$ (light blue). The cerebellum generates cortical feedback predictions $\hat{\delta}_3$ (blue) given cortical activity $h_3$ (black), and learns using inferior olive (diamond) error signals (red arrow). Before $t_3$, cortical feedback is not readily available, thus the cerebellum learns through bootstrapping. In this case the inferior olive (diamond) compares the old cerebellar prediction (e.g. $\hat{\delta}_2$) with the new one (e.g. $\hat{\delta}_3$) to generate cerebellar learning signals (red arrow; see main text for details).

there is a considerable divergence from the $\sim 250$ million input mossy fibres. To replicate this, in our experiments we include a high divergence ratio ($1 : 4\text{-}5$) between the RNN network size and the hidden layer of the cerebellar network (see SM for network sizes). In our experiments this ratio helped the cerebellar module learn more quickly (not shown).

At the functional level, our model suggests that the cerebellum should encode a variable dependent on error signals (in particular, the error gradient with respect to neocortical activity), consistent with neural recordings showing cerebellar-encoding of kinematic errors during motor learning [27–29], but also reward prediction signals in cognitive tasks [30].

Finally, we conform to the classical view that the inferior olive would be the site at which cerebellar prediction and feedback are compared to compute the cerebellar cost $\mathcal{L}_\mathcal{C}$ which drives cerebellar learning [7–9, 31]. Crucially, we also highlight the cerebellum's ability to learn in the case of delayed feedback - which is necessarily the case under this model - via its specially adapted timing rules for plasticity at the parallel fibre synapse [32].

### 2.1.1 Relationship to existing cerebellar computational models

For over half a century computational neuroscientists have posited that the feedforward circuitry and available error signals (stemming from the inferior olive) make the cerebellum well suited for pattern recognition [7–9, 33, 34]. This computational perspective is in line with our model, but which patterns should be learnt, and how is cerebellar output used by the rest of the brain has remained unclear. The most prevalent theory points towards the cerebellum as an "internal model" of the nervous system [10], which may arise in two forms, an *inverse* model or a *forward* model.

We argue that ccRNN is in fact an instance of the forward model hypothesis (see SM A for a link between DNIs and the inverse model). In the classical forward model of sensorimotor control, the cerebellum receives an efferent copy of the motor command from the motor cortex [11, 35], and the

respective sensory feedback. With these two inputs the forward model learns to predict the sensory consequences of motor commands. Generalising to the non-motor domain, a similar predictive model can be applied to the brain activity between virtually *any* two brain areas [36]. The prefrontal cortex and the temporo-parietal cortex, for instance, both involved in planning of cognitive behaviour and decision making, are possible cognitive-associated areas where a forward "mental" model may be applied [12–14, 36]. Our model also takes neural activity $h$ from *any* brain area and its associated feedback signals $\delta$, which provides, to the best of our knowledge, the first explicit computational implementation of postulated brain-wide forward models. In addition, existing computational models have so far only been used in simplistic sensorimotor tasks, whereas our model can be readily applied to a wide range of tasks as we demonstrate below.

### 2.1.2 Encoding feedback gradients in the brain

Our systems level model uses BPTT to generate temporal feedback signals. We use weak forms of truncated-BPTT (i.e. with a reduced BPTT window), using only one BPTT step in some cases, which mimics a more biologically realistic setting. It is in principle possible to use models capable of generating biologically plausible feedback [6, 37–41]. Computational models of spatial back-propagation of gradients (i.e. across layers) suggest that gradient-encoding feedback signals must be encoded in distal dendrites [37, 38]. In addition, these models demonstrate that explicit gradient feedback information is not needed, but rather that this can be reconstructed locally in dendritic microcircuits using cortical feedback [38]. The cerebellum provides feedback connections via higher order thalamocortical loops [42] that target apical dendrites of pyramidal cells in the neocortex [43–45]. These cerebellar feedback loops thus provide a good substrate for driving cortical learning through cortical feedback prediction, which is consistent with our model. On the other hand Bellec et al. [46] have shown that temporal gradients as used in BPTT can be approximated by biologically plausible eligibility traces that transmit gradient information forward in time. Both of these solutions can, in principle, be incorporated into our framework, in which ccRNN would predict feedback activity originating from upstream brain areas (as in Sacramento et al. [38]) and/or make forward predictions which are then integrated with locally computed eligibility traces [46].

## 3 Results

We contrast a model with cerebellar feedback by comparing ccRNN to a purely cortical - that is, without cerebellar predicted feedback - RNN (cRNN) on a variety of tasks. Tasks are chosen so as to broadly approximate conditions in which cerebellar patients have shown deficits, including simple and then more complex sensorimotor control/planning paradigms before ending with a challenging language-based task.

Cortical recurrent neural networks are modelled as long short-term memory networks (LSTMs) [47] in line with previous work [48, 49]. A linear readout is added on top of the RNN to compute final model output. The window of cortical temporal feedback is modelled using BPTT of a specific truncation window. We refer to feedback originating from the cerebellum as *predicted feedback*.

For the sensorimotor tasks we also test the models with different external feedback intervals. This defines how often the model has access to an external teaching signal (e.g. only every 2 timesteps). This mirrors a biologically-relevant setting where visual feedback is not continually available but can only be sampled at some rate. Therefore, the model is trained with external feedback only for some timesteps, and that it should ideally generalise to the remaining timesteps. We also evaluated the model performance under these testing conditions (i.e. in which external feedback was not available) to yield an *ataxia score*, a measure designed to quantify the irregularity of movement associated with cerebellar ataxia [50]. Concretely, we compute the ataxia score as the total deviation from the model output to the targets provided during training *as well as* those for which external feedback was unavailable.

$$\text{ataxia score} = \sum_{t \in \text{FB}} \mathcal{L}\left(y_t, \hat{y}_t\right) + \sum_{t \notin \text{FB}} \mathcal{L}\left(y_t, \hat{y}_t\right) \qquad (2)$$

where $y_t$ and $\hat{y}_t$ denotes the target and model output at time $t$ of the sequence, respectively, $\mathcal{L}$ is the task-associated error function (e.g. mean squared error), and FB denotes the set of times at which error feedback is given. Note that the first term is simply the training error (which defines the model optimisation problem), whereas the second term quantifies the model's ability to generalise to unseen parts of the data (for example, intermediate points on a line).

## 3.1 Simple sensorimotor task: line drawing

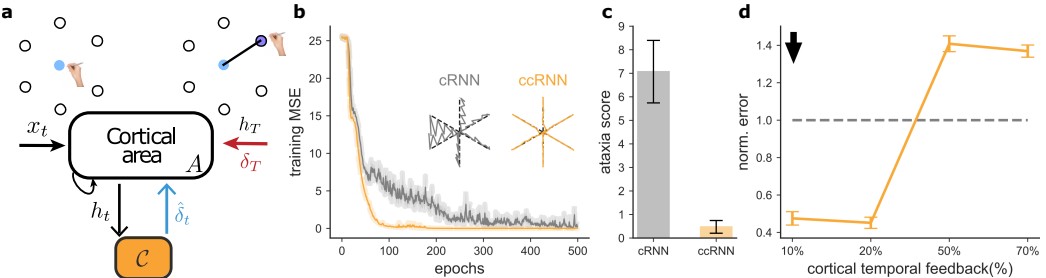

Figure 2: Line drawing task. (**a**) Schematic of the task. One out of six cues is given at $t_1$ and the network must learn to draw a straight line to the respective target. Sparse feedback is used so that an explicit target is only provided at every other timestep during training. (**b**) Learning curves for cortical RNN (cRNN; grey) and cortico-cerebellar RNN (ccRNN; orange). These networks use cortical temporal feedback of $10\%$ (cf. d), increasing the difficulty of learning in the RNN. Trajectories produced by the models towards 6 targets are given as insets. (**c**) Ataxia score for both models at the end of training. (**d**) Mean squared error of the ccRNN normalised to that of cRNN over different cortical temporal feedback windows (reported as a percentage of total task length); arrow indicates level of feedback used in (b,c). All experiments used 10 seeds.

Inspired by classical sensorimotor studies in the cerebellum we first test a simple line drawing task [23–25], in which given a cue at time $t_1$ the network needs to produce a straight line towards one of 6 targets over 10 timesteps (Fig. 2a; see SM for more details). In order to model a more biologically realistic setting the external feedback (i.e. how far the model output is from a straight line) is only provided at every other timestep (sparse external feedback). Although the cRNN model learns to perform the task it does so more slowly and with higher variability (Fig. 2b), in line with cerebellar experiments [23–25]. Importantly, we also observe differences in the model output (Fig. 2b). Even though the models were only trained with sparse external feedback the ccRNN generalises to a near-perfect straight line whereas the cRNN produces a much more irregular, oscillatory-like output (Fig. 2b), leading to an increased ataxia score (Fig. 2c). Moreover, our model predicts that the cerebellum module is beneficial when the main cortical recurrent network struggles to learn the task on its own (i.e. when using a short temporal feedback window; Fig. 2d, S1). This is consistent with the observation that the cerebellum is more important to correct for larger errors during a motor task [51]. In addition to this prediction our model makes several other predictions from a systems level down to a cellular level. Below we highlight some of these predictions using the line drawing task and relate it to the experimental literature where possible.

**Facilitation of learning depends on feedback interval** The benefits provided by the ccRNN model are stronger at intermediate feedback intervals (Fig. 3a). In particular, we find that when the external feedback is given continuously there are only minor benefits of having the cerebellar module, but if the external feedback interval is increased then having a predictive cerebellar module becomes more important until the feedback interval is just too large for the models to be able to learn the task (Fig. 3a). We observe these differences because the feedback predicted by the cerebellum is only useful once the external feedback is relatively infrequent. Although this is broadly in line with the important role of feedback in cerebellar-mediated learning [23, 52, 53] this exact prediction remains to be tested.

**Cortico-cerebellar coupling is training-dependent** As a measure of the cortico-cerebellar coupling we calculated the pairwise correlations between the activity of each neuron in the main cortical RNN and the activity of each hidden neuron (granule cell) in the cerebellar module (see SM). Our model reveals two distinct phases of cortico-cerebellar coupling during learning (Fig. 3b). During the fast phase of learning we observe a noticeable rise in the average cortico-cerebellar coupling (in line with [13]). In addition, our model predicts that as training continues the population correlation should gradually decrease. In the model these changes in coupling are explained by the fact that initially the model needs to output relatively high feedback due to the high errors, but as learning progresses

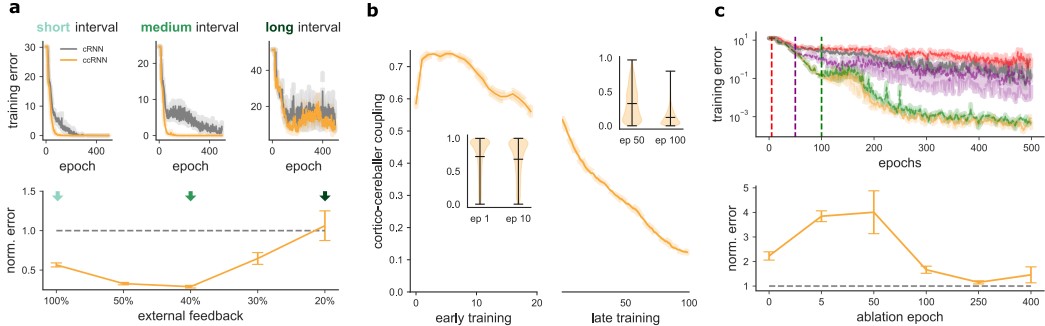

Figure 3: Model predictions using simple line drawing task. (**a**) Cerebellar feedback facilitates learning across a range external feedback intervals. (**b**) Cortico-cerebellar coupling measured as pairwise correlations for first 100 epochs of learning. Insets show the change in distribution of correlations at respective epochs (left: early epochs; right: later epochs). (**c**) Importance of cerebellar feedback reduces over learning revealed by ablation study. Top: Learning curves for a few example points of ablation. Vertical lines represent points of ablation. Bottom: Summary plot of the training error (ccRNN normalised to cRNN) across epoch of ablation. Error bars represent standard error of the mean using 10 seeds.

errors (and feedback) becomes gradually smaller, thus making the cerebellar module less correlated with the activity of the main cortical network.

**Importance of cerebellar feedback reduces over learning**    The decrease in coupling over learning predicts that the importance of the cerebellar feedback should become weaker over learning. To test this idea and which phases of learning are more crucial for the cerebellar-mediated facilitation of learning we completely ablated the cerebellar module at different points during learning (Fig. 3c). Although these results are consistent with the classical observations made in cerebellar patients [23–25] this ablation study predicts a nonlinear relationship between ablation time and performance impairment, which has not been tested experimentally. Our model predicts that ablating the cerebellum after learning has started actually has a more detrimental impact than if there was not a cerebellum to start with (Fig. 3c). This happens because when a cerebellum module is present the main cortical network starts learning by relying partially on the feedback predicted by the cerebellum, so that if this component is suddenly removed the learning trajectory of the main network is perturbed and needs to be readjusted. After this first critical phase of learning the cerebellum becomes gradually less important ($> 50$ epochs). These results echo the existing literature [54, 55] in that though shared cortico-cerebellar dynamics might emerge during the acquisition of a novel motor sequence, these dynamics can become less interdependent once knowledge becomes consolidated.

## 3.2    Advanced sensorimotor tasks

To test how the previous line drawing task generalises to a more realistic setting with continuous input we introduce two sensorimotor tasks which build on the classical MNIST dataset [56]. Here we present the model with a given MNIST image sequentially row by row (28 rows, each with 28 pixels), whilst training the model to simultaneously draw a digit-specific (i) straight line (ld-seqMNIST) or (ii) a digit template (dd-seqMNIST) (Fig. 4a). The latter case is inspired directly from known cases of cerebellar agraphia [57]. As before we employ sparse external feedback at a certain interval (every 4 timesteps).

As in the line drawing task ccRNN learns faster than cRNN while also showing a reduced ataxia score (Fig. 4b, c). The positive effect of cerebellar feedback is less apparent when the model was trained to write digits, possibly due to the less predictable, non-linear nature of target output and subsequently external feedback. As with the line drawing task we find that predicted feedback is most valuable where strong constraints on BPTT are enforced (Figs. 4d, S3). We observed a similar dependency to the simple line drawing task on the level of external feedback (Fig. S4), cortico-cerebral coupling (Fig. S5), and ablations (Fig. S6). However, in contrast to the simple line drawing task, cerebellar feedback is still relatively important even in later stages of learning. This is due to the non-zero error even after convergence in these tasks.

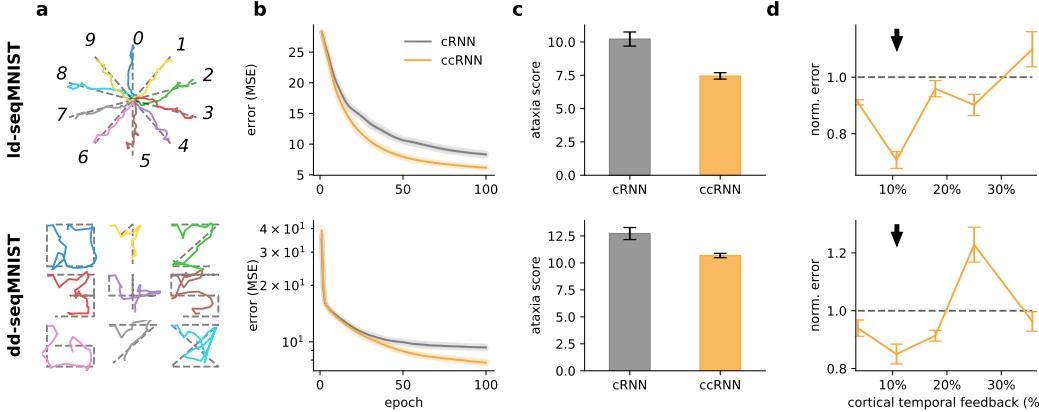

Figure 4: Advanced sensorimotor tasks based on MNIST dataset. (**a**) Example model output given an MNIST digit as input (colour-coded by digit) from a trained ccRNN model for the line drawing sequential MNIST task (ld-seqMNIST, top) and the digit drawing sequential MNIST (dd-seqMNIST, bottom); underlying grey dots represent target output. Sparse feedback is used so that an explicit target is only provided at every 4 timesteps during training. (**b**) Training mean-squared error. **c** Ataxia score for both models at end of training. (**d**) ccRNN error (normalised to cRNN error) over different cortical temporal feedback windows (reported as a percentage of total task length); arrow indicates level of feedback used in (a,b,c)). Error bars represent standard error of the mean using 10 seeds.

Finally, we also considered how ccRNN performs on the more standard sequential MNIST task, in which it is necessary to classify the digit at the end of the sequence [58]. In line with the regression tasks, we find significantly faster learning ( $10\%$ higher accuracy after about 10 epochs) when using the ccRNN model (Fig. S2). This is consistent with a role of the cerebellum in decision making or sensory discrimination tasks [21, 59, 60].

## 3.3 Cognitive task: Caption generation

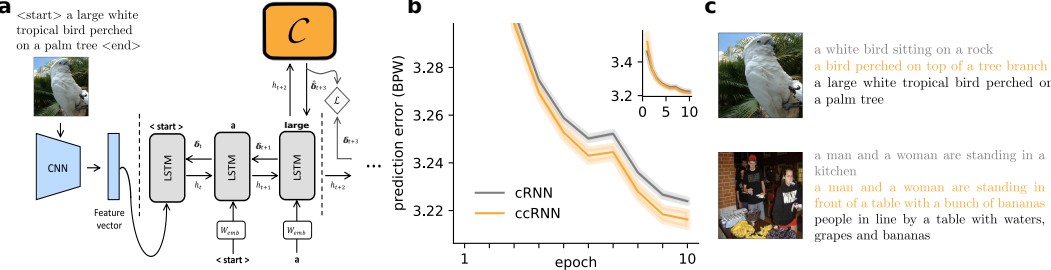

Figure 5: Caption generation task. **a** Model schematic with CNN (blue), cortical RNN (LSTM in gray) and cerebellar module (orange). CNN is pretrained on a image dataset, while the RNN is trained to predict the next word in a caption. **b** Learning curves in bits per word (BPW). **c** Sample test image with associated model produced captions (colour coded as in **b**; black denotes gold standard caption). More examples are given in the SM. Error bars represent standard error of the mean using 5 seeds.

We emphasise that our framework does not only apply to sensorimotor tasks, but should generalise to virtually any task and learning paradigm (supervised, unsupervised and reinforcement learning). To demonstrate this and inspired by cognitive tasks in which cerebellar patients have shown deficits [61] we test our models in a caption generation task. In this task the network learns directly from the data (i.e. unsupervised learning) to generate a textual description for a given image. All models have two components: a pretrained convolutional neural network (CNN) to extract a lower dimensional representation of the image, and an RNN to learn a simple model of language given a visual representation from the CNN (Fig. 5a).

We use a standard dataset (ILSVRC-2012-CLS; Russakovsky et al. [62]) and the networks are trained to maximise the likelihood of each word given an image (SM for more details). We find that also in this more challeging task the cerebellum module manages to learn good enough feedback to improve learning when compared with cRNN (Fig. 5b). Interestingly, the captions generated on test images are qualitatively more accurate for the ccRNN model (Figs. 5c, S8), suggesting that the ccRNN model better captures the contextual nature of this task, in line with experimental observations [61]. Moreover, ccRNN also outputs a better model of language as assessed by language metrics (Table S2).

## 4    Conclusions and discussion

We have introduced a systems level model [5] of cortico-cerebellar function, where the cerebellum's role is to provide the neocortex with predicted temporal feedback. We propose that an existing deep learning framework can mimic the function of one of the largest projection pathways in the brain, cortico-cerebellar networks. This systems level framework abides to the classical notion of the cerebellum as a pattern learning multi-layer perceptron in the context of the forward model hypothesis. In particular, our model suggests that the cerebellum encodes future errors (or analogously, rewards) which are then transmitted to the neocortex for efficient learning. It thus proposes how the brain might learn efficiently from future teaching signals: a key requirement of temporal credit assignment.

One of the advantages of our model (ccRNN) is that it can be readily applied to a wide range of tasks as a systems-level model of cortico-cerebellar interactions. Indeed, our model makes explicit the concept of a "cognitive error" which is directly connected to growing empirical evidence for a cerebellar role in working memory, planning, language and beyond [60, 63]. However, to make more specific predictions at the cellular, sub-cellular and cell-type level an important next step is model cortical feedback errors in a more biologically plausible fashion [38, 64].

Our model makes a number of predictions (see above). For example, that cerebellar involvement is most beneficial where neocortical temporal learning mechanisms are not long enough for the task at hand. This offers an explanation for the often detrimental impact cerebellar impairments can have on motor learning which are often temporally challenging. It is important to highlight that our results do not predict that cerebellar impairments will necessarily lead to an inability to learn, but merely a slower learning curve and perhaps lower performance threshold. There are however specific conditions in which the cerebellum is optimally placed to facilitate learning according to our model, namely the regularity of the external feedback, the ability for learning of cortical networks, the difficulty of the task and the exact phase of learning. In addition our model also makes predictions about cortico-cerebellar coupling, predicting a steady decrease of the population correlation as learning progresses.

Our model puts forward a solution for how the brain deals with delayed feedback signals, by relying on the cerebellum to predict future feedback signals. A key experiment which could be used to test our model would be to demonstrate that brain areas responsible for computing prediction errors are not by themselves a requisite for plasticity. The cerebellum by itself should be sufficient for cortical learning, provided that it has learnt a good predictive model of future feedback.

There are a number of interesting directions to take forward with this model. For example, in the present study we do not explicitly model the long-range projections via the thalamus and pons that mediate cortico-cerebellar interactions [43–45], but for simplicity we have assumed direct connectivity. We predict that introducing these intermediate structures with low dimensionality has two potential important functions: (i) force the cerebellum to focus on learning only the most important feedback signals and (ii) gate in and out cerebellar feedback so that this is only allowed to modulate learning when beneficial. Moreover, here we have focused on a variant of the model that predicts feedback signals needed for learning, but it is likely that the cerebellum is also involved in speeding up feedforward processing as in forward DNIs (see SM).

On the deep learning side there are also a number of promising options to explore. Predicting feedback is a hard task, due to its continuously changing nature. There are a number of architectural and cellular features of the cerebellum that may make learning feedback predictions more efficient in these models, namely mossy sparse connectivity (as observed between mossy fibres and granule cells [65, 66]) and modularity (as observed throughout the cerebellum [67]). At the same time, it may also be fruitful to consider decoupling neural processing more generally. Various other, potentially

non-cerebellar, methods for unlocking have been proposed in machine learning that may provide interesting avenues for neuroscience [68–70].

Overall, ccRNN provides a natural link between deep learning and existing cortico-cerebellar ideas and more classical cerebellar models. Moving forward we hope the framework to offer a novel but concrete framework with which to study the cerebro-cerebellar loop for neuroscientists and as a source of inspiration for machine learners.

## Acknowledgments and Disclosure of Funding

We would like to thank the Neural & Machine Learning group, Paul Anastasiades, Paul Chadderton and Paul Dodson for useful discussions. JP was funded by a EPSRC Doctoral Training Partnership award and EB by the Wellcome Trust (Neural Dynamics PhD Program). This work made use of the HPC system Blue Pebble at the University of Bristol, UK.

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
