# A Forward DNI

In this paper we have focused on the backward (or feedback) DNI, but there is another interesting paradigm between two neural networks dubbed "forward" DNI. Here we describe this variant of the model and below its link to the cerebellum.

The architecture remains the same; that is, we have a feedforward or recurrent main network and a separate feedforward network (synthesiser; $\mathcal{C}^{forward}$) to which it forms a loop. The difference to backward DNI is that now the synthesiser predicts forward activity, not backward. Specifically, we have $\mathcal{C}^{forward}(h_i) = \hat{h}_j \approx h_j$, where $h_i$ is the activity of the main network at layer (or time) $i$ and $h_j$ with $j > i$ is the activity at some later layer (or time). As with backward DNI we constantly update the synthesiser parameters based on its difference to its target, in this case $\mathcal{L}_{\mathcal{C}^{forward}} = ||\hat{h}_j - h_j||$.

Though more nuanced, the goal of forward DNI as presented in [5] is also to hasten learning. As an example, suppose we have a feedforward network as the main model and equip a backward synthesiser (one which predicts same layer error gradients) at each layer as well as a forward synthesiser which projects from the original network input $x$ onto each layer. The result is that given $x$, there are only two stages of processing before the parameters of layer $j$, for *any* $j > 0$, can be updated: approximating $h_j$ with $\mathcal{C}^{forward}(x) = \hat{h}_j$, then apply the backward synthesiser $\mathcal{C}(\hat{h}_j) \approx \frac{dL}{dh_j}$. In this case, we have what Jaderberg et al. dub as a "full unlock" of the forward and backward pass.

## A.1 Relationship to cerebellum

As with backward DNI we can frame forward DNI as an instance of the internal model hypothesis [10]. Now, however, we seek an analogy to the inverse model hypothesis. Under the classical formulation of the inverse model, the cerebellum receives the current and desired sensory state before issuing a motor command; i.e. the cerebellum now acts as the *controller*. An analogous model can be derived in the cognitive domain, where the cerebellum might learn to manipulate some given "instruction" (desired/current state) in a manner which approximates the prefrontal cortex in its expression of a "mental model" (controlled object) [36].

If we consider the spatial case of forward DNI and interpret the layers between which $\mathcal{C}^{forward}$ approximates as, for example, the motor or prefrontal cortex, then the link towards the inverse model becomes clear. In both schemes, the cerebellum receives initial states upstream (instructions) and learns to mimic the forward computations which then take place in the neocortex. We also point out that though the temporal case of forward DNI was not originally considered in [5], there remain clear analogies to proposed cerebellar function. In fact, it was recently suggested that the cerebellum mimics motor processing over several timesteps (Fig. 7 in [71]), exactly analogous to temporal backward DNI where the main model is an motor-associated RNN.

In general, the likeness in formulation between DNI and the cerebellar internal model hypothesis can be summarised in table S1.

| | Forward Model | Backward DNI | Inverse Model | Forward DNI |
|---|---|---|---|---|
| Controller | Neocortex | Main model | Cerebellum | Synthesiser |
| Input | $\mathbf{h}_m$ | $\mathbf{h}_m$ | $\mathbf{h}_s, \mathbf{d}_s$ | $\mathbf{h}_s$ |
| Output | $f_{\text{unk}}(\mathbf{h}_m)$ | $\mathcal{C}^B(\mathbf{h}_m) = \frac{\partial \hat{L}}{\partial \mathbf{h}_m}$ | $f_{\text{unk}}(\mathbf{h}_s, \mathbf{d}_s)$ | $\mathcal{C}^F(\mathbf{h}_s)$ |
| Ouput destination | Neocortex | Main model | Controlled object | Main model |

Table S1: Relationship of the internal models of the cerebellum with DNIs. The properties of the forward model of the cerebellum can be set against those of backward DNI (blue); similarly, the properties of the inverse model of the cerebellum can be set against those of forward DNI (red). Notation is largely consistent with section 2 of the main text: $\mathbf{h}_m, \mathbf{h}_s$ denotes the hidden activity of a motor area and sensory area, respectively; $\mathcal{C}^B, \mathcal{C}^F$ denotes the computation of backward DNI and forward DNI, respectively; $L$ denotes the loss function. In addition, the inverse model of the cerebellum traditionally also has access to a *desired* state $\mathbf{d}_s$ (in particular, one can consider this a special case of "context" provided to the synthesiser; cf [5]). There are no explicit equations for the computational processes of the forward and inverse models, and both are thus represented by the unknown function $f_{\text{unk}}$.

# B   Experimental details

In each of our tasks we use a long-short term memory network (LSTM; [47]) as the main "cortical" network [48], and a simple feedforward network of one (sensorimotor tasks) or two hidden layers (caption generation) as the synthesiser "cerebellar" network. As in [5] the cerebellar network predicts gradients for both the memory cell and output state of the LSTM, so that the cerebellar input and output size is two times the number of LSTM units; furthermore, the synthetic gradient is scaled by a factor of 0.1 before being used by the main model for stability purposes. The final readout of the model is a (trained) linear sum of the LSTM output states. All networks are optimised using ADAM [72] (see learning rates below).

In each experiment all initial LSTM parameters are drawn from the uniform distribution $\mathcal{U}(\frac{1}{\sqrt{n_{\text{LSTM}}}}, \frac{1}{\sqrt{n_{\text{LSTM}}}})$, where $n_{\text{LSTM}}$ is the number of LSTM units. The weights of the readout network and the feedforward weights of the cerebellar network (other than the final layer) are initialised according to $\mathcal{U}(-b_k, b_k)$ where $b_k$ denotes the "kaiming bound" as computed in [73] (slope $a = \sqrt{5}$), and the biases are draw from $\mathcal{U}(\frac{1}{\sqrt{n_{\text{inp}}}}, \frac{1}{\sqrt{n_{\text{inp}}}})$, where $n_{\text{inp}}$ denotes the input size of the layer. As in [5], the last layer (both weights and bias) of the cerebellar network is zero-initialised, so that the produced synthetic gradients at the start are zero.

During learning, backpropagation through time (BPTT) takes place strictly within distinct time windows (truncations), and computed gradients are not propagated between them: *truncated* BPTT. We split the original input into truncations as follows. Given an input sequence of $N$ timesteps $x_1, x_2, \ldots, x_N$ and a truncation size $T$, we divide the sequence into $T$ sized truncations with any remainder going to the last truncation. In other words, the sequence is now made up truncations of $(x_1, \ldots, x_T), (x_{T+1}, \ldots, x_{2T}), \ldots, (x_{(m-1)T+1}, \ldots, x_{mT}), (x_{N-r}, \ldots, x_N)$, where $N = mT + r$ for positive integers $m, r$ with $0 \le r < T$. Note that, along with the value $T$, how well the sequence is divided into truncations (i.e. values $m, r$) is an important factor for learning and can cause noticeable non-linearities (Fig. 2d, 4d). For the line drawing and seqMNIST based tasks where there might be truncation windows where the error gradient is purely synthetic, we accumulate error gradients and apply gradient descent strictly at the end of the sequence. This is to ensure that for each model there is the same overall number of weight updates, a potentially important detail for fairness when using optimisation tools such as ADAM. For the image captioning task, where there is sure to be an error signal at every truncation, we update the model weights as soon as the error gradients become available.

In the line drawing and seqMNIST based tasks, to test the effect of predicted feedback against the availability of "organic" error signals which occur at any timestep where a target is provided, we vary the *external feedback interval*. Given feedback interval $n$, the target is only available every $n$ timesteps. An arguable but helpful analogy might be the rate at which one receives visual information whilst performing a task (e.g. drawing freehand).

In general, (sensible) hyperparameters were selected by hand after a few trial runs. We used the PyTorch library for all neural network models. In particular, our DNI implementation is based on code available at `github.com/koz4k/dni-pytorch`. The code used for our experiments is available at `https://github.com/neuralml/ccDNI`.

**Normalised error**   To calculate the normalised error with respect to a given model (Figs. 2d, 3a, c, 4d, S4) we take the ratio of total errors during learning (all epochs). For example, the normalised error of ccRNN with respect to cRNN is $\frac{\text{error(ccRNN)}}{\text{error(cRNN)}}$. Note that in the ablation case we compare against an unaffected ccRNN and only consider the respective errors post-ablation. e.g. the normalised error for a model with cerebellar ablation at epoch 50 is $\frac{\text{error(ablated)}_{>50}}{\text{error(ccRNN)}_{>50}}$.

**Cortico-cerebellar coupling**   To analyse how the coupling between the two model components changes over learning (inspired by [74]) we consider the (absolute) Pearson correlation between a given LSTM unit (both cell and output states) and a given unit in the cerebellar hidden (granular) layer over different bins during training. Values presented are the average over all RNN/cerebellar hidden unit pairs.

**Computing details**  All experiments were conducted on the BluePebble super computer at the university of Bristol; mostly on GPUs (GeForce RTX 2080 Ti) and some on CPUs (Intel(R) Xeon(R) Silver 4112 CPU @ 2.60GHz). We estimate the total compute time (including unobserved results) to be in the order of $\sim 400$ hours.

## B.1  Line drawing task

In the line drawing task, an LSTM network receives a discrete input cue which signals the network to either 1. stay at zero or 2. move across an associated line (equally spaced points) in 2D space over a period of 10 timesteps. Here we set 6 distinct non-zero input-target pairs $\{(x_i, y_i)\}_{i=1}^6$, where each input $x_i$ is a (one dimensional) integer $\in \{\pm 1, \pm 2, \pm 3\}$, $y_1 = 0$ throughout, and the remaining targets $\{y_i\}_{i=2}^6$ are lines whose end points lie equidistantly on a circle centred on the origin with radius 10. Once an input cue is received at timestep $t_0$, the model receives no new information (i.e. all future input is zero). The model is trained to minimise the mean squared error (MSE) between its output and the cue-based target.

The cortical network has one hidden layer of 50 LSTM units and the cerebellar network contains one hidden layer of 400 neurons. The initial learning rate is set to 0.001. Each epoch comprises 20 batches of 50 randomised examples. Unless explicitly stated we use a truncation size of $T = 1$ which covers $10\%$ of the total task duration. Model results are averaged over 10 random seeds (with error bars), where each seed determines the initial weights of the network.

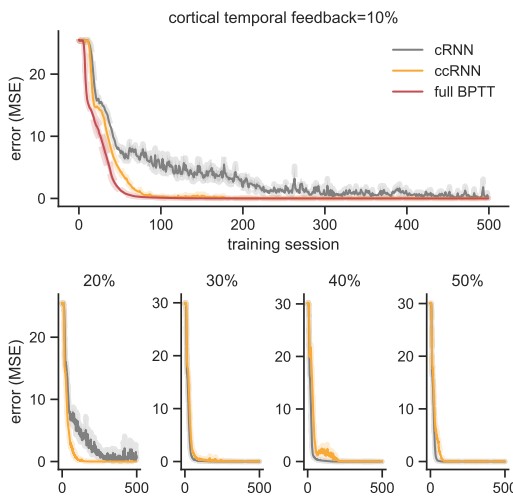

Figure S1: Learning under different cortical temporal feedback windows intervals for the linedrawing task (cf Fig. 2d). Size of window is presented as percentage of task duration (10 timesteps). Results presented in main text (Fig. 2b) shown on top row along with RNN trained with full backpropagation through time (i.e. cortical temporal feedback = 100%).

## B.2  Sequential MNIST tasks

For each seqMNIST based task the model receives the same temporal MNIST input, and the tasks are only differentiated by the model output. Given a $28 \times 28$ MNIST image, at timestep $i$ the model receives the pixels from row $i$ of the image, so that there are 28 total timesteps and an input size of 28.

In each case we have one hidden layer of 30 LSTM units in the main model and one hidden layer of 300 hidden units in the feedforward cerebellar network. Data was presented in batches of 50 with an initial learning rate of 0.0001.

Training and validation data was assigned a $4 : 1$ split, containing 48000 and 12000 distinct image/number pairs respectively. Unless explicitly stated, the truncation value was $T = 3$ which is $\sim 10\%$ of the task duration. Model results in main text (Fig. 4) taken over 10 random seeds for weight initialisation (3 seeds for extra supplementary figures).

### B.2.1  ld-seqMNIST

In this variant each number 0-9 MNIST image is allocated an associated $xy$ position on the edge of a circle centred at 0 with radius 10, and must follow a line (equally spaced points) towards that position (Fig. 4a, top). With the model output then a vector of size 2, the training loss is defined at the end by the mean squared error (MSE) between the output of the model and the points forming the target line.

### B.2.2  dd-seqMNIST

Like ld-seqMNIST, in this variant the model outputs a sequence of 2D coordinates corresponding the given image. The target sequence however is now of a highly non-linear form, and in this case actually resembles the shape of the number itself (Fig. 4a, bottom; number 9 not shown). The model is then trained at each timestep to minimise the MSE between the model output and that target shape.

For each number, the corresponding target drawing lies in $[0, 1]^2$, with the gap between each successive point roughly the same. To prevent the model being too harshly judged at timestep 1, all drawings begin in the top left corner $(0, 1)$ (apart from the drawing of 1 which begins slightly beneath/to the right). MSE scores are reported as 100 times their raw values to ease comparison with ld-seqMNIST.

### B.2.3  SeqMNIST

This is the standard form of seqMNIST (see Fig. S2) and used as a case of a discrimination (or decision making) task and one at which the target is hardly available to the model (only at the end). In this case at the end of the presentation of the image the model must classify in the image as a number between 0 and 9. The output of the model is a vector probabilities of size 10 (one entry for each number), and the model was trained to maximise the likelihood of the correct number.

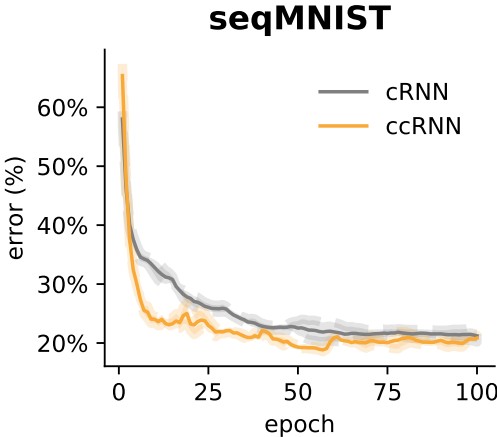

Figure S2: Learning curve using the validation set across epochs for the sequential MNIST task. Truncation window $T = 3$ ($\sim 10\%$ of sequence length) applied.

### B.3  Caption Generation

The architecture for the caption generation task consists of a pretrained convolutional neural network (CNN) coupled with an RNN (LSTM). The synthesiser (cerebellar network) only communicates with the LSTM. The LSTM network has one layer of 256 LSTM units and the cerebellar network has two hidden layers of 1024 neurons.

The dynamics from image to caption is as follows. As part of image preprocessing and data augmentation, a given image is randomly cropped to size $224 \times 224$, flipped horizontally with even chance, and appropriately normalised to be given to a pretrained Resnet model [75]. A feature vector $X$ of size 256 is thus obtained and is passed to the LSTM at timestep 0. The LSTM is subsequently presented the "gold standard" caption $\{w_i\}_{i=1}^n$ one word per timestep, each time learning to predict the next word; i.e., at timestep $t$ the model learns $P(w_t|X, \{w_i\}_{i=1}^{t-1})$. The network simultaneously learns a word embedding so that each word $w_i$ is first transformed to a feature vector of size 256

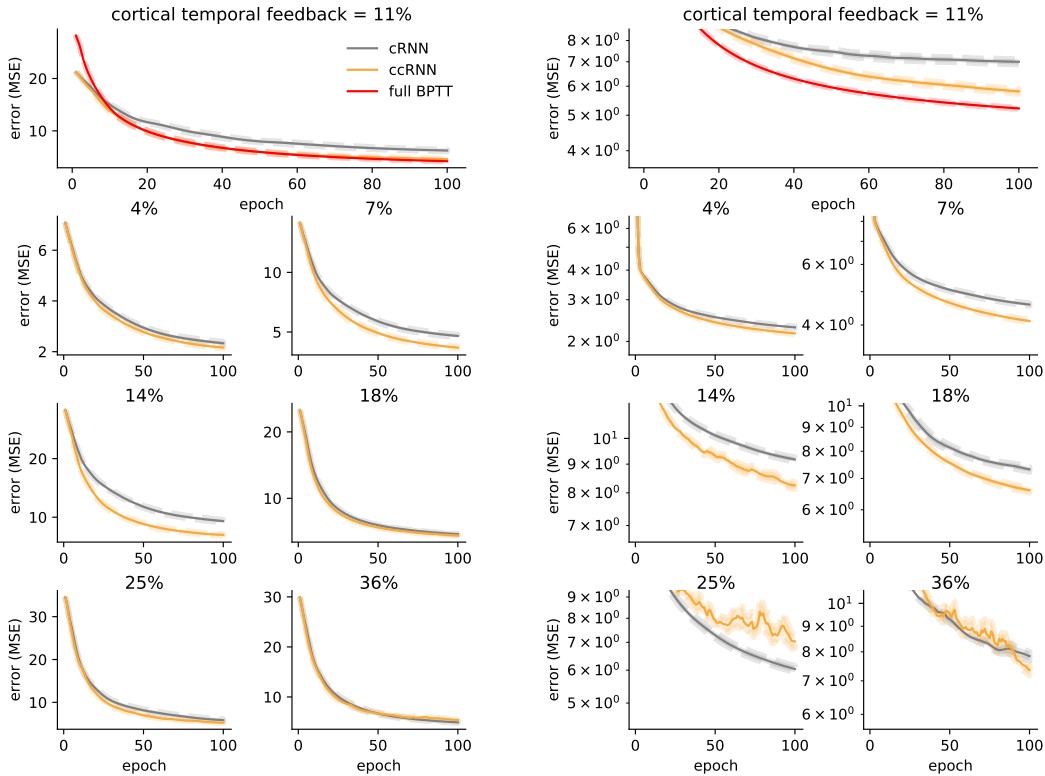

Figure S3: Learning under different cortical temporal feedback windows intervals for the ld-seqMNIST (left) and dd-seqMNIST (right) tasks (cf Fig. 4d). Size of window is presented as percentage of task duration (28 timesteps). Results presented in main text (Fig. 4b) shown on top row along with RNN trained with full backpropagation through time (i.e. cortical temporal feedback = 100%).

before being served as input. With a preset vocabulary of 9956 distinct words, the final output of the model ($P(w_i)$) is a probability vector of size 9956.

We found the models to be generally prone to overfitting the training data. For this reason, we apply dropout (during training) on the input to the LSTM, where a given input element is set to zero with $p = 0.5$ probability.

Once training is complete the models can generate their own unique captions to previously unseen images (Figs. 5, S8). Given an image at timestep 0, the model applies a 'greedy search' where the output of the model at timestep $i$ is the word with the highest probability, and the same word is then provided as input to the model at timestep $i + 1$. In this way the model can autonomously output an entire sequence of words which forms a predicted caption. In the (highly) rare case where the model generates a sequence of $> 20$ words, we consider only the first 20 words as its caption.

The coco training set ILSVRC-2012-CLS [62] holds 414113 total image-caption pairs with 82783 unique images while the held-out validation set (used for Fig 5b, c) holds 202654 with 40504 unique images; note that each image therefore has $\sim 5$ distinct gold standard captions. Training takes place in batches of 100 image/caption pairs, with an initial learning rate of 0.001. Model performance averaged (with error bars) over 5 random seeds for weight initialisation.

In order to judge the models beyond their learning curves in bits per word (BPW), we quantify their ability to generate captions using a variety of language modelling metrics popular in the realm of language evaluation (image captioning, machine translation, etc). In particular, we compare model-generated captions against the gold standard captions using standard metrics in language modeling (Table S2).

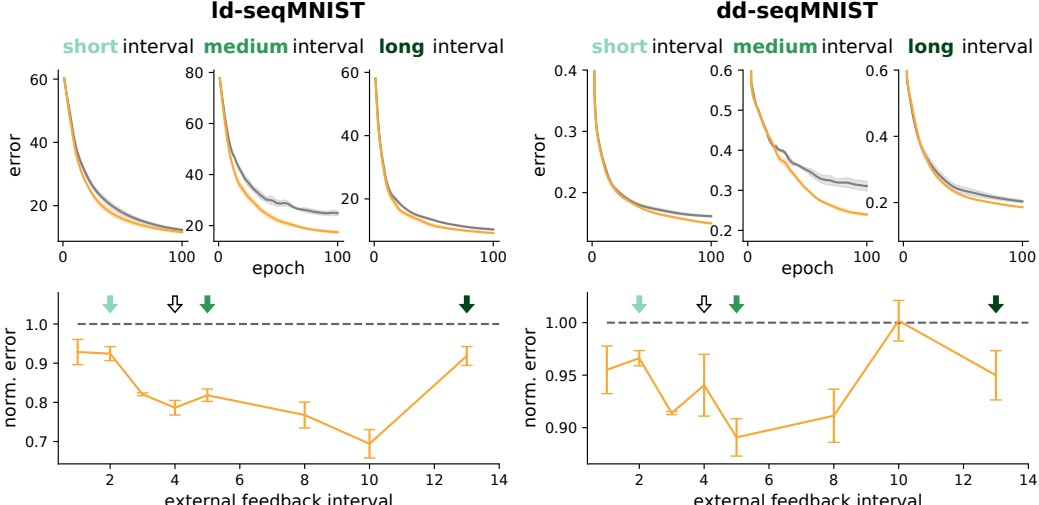

Figure S4: Learning under different external feedback intervals for the ld-seqMNIST (left) and dd-seqMNIST (right) tasks. Top row: training curves with different feedback intervals. Bottom row: ccRNN error normalised to cRNN across different feedback intervals. Opaque arrows colour coded to the text above. Transparent arrow designates the feedback interval used for experiments presented in main text.

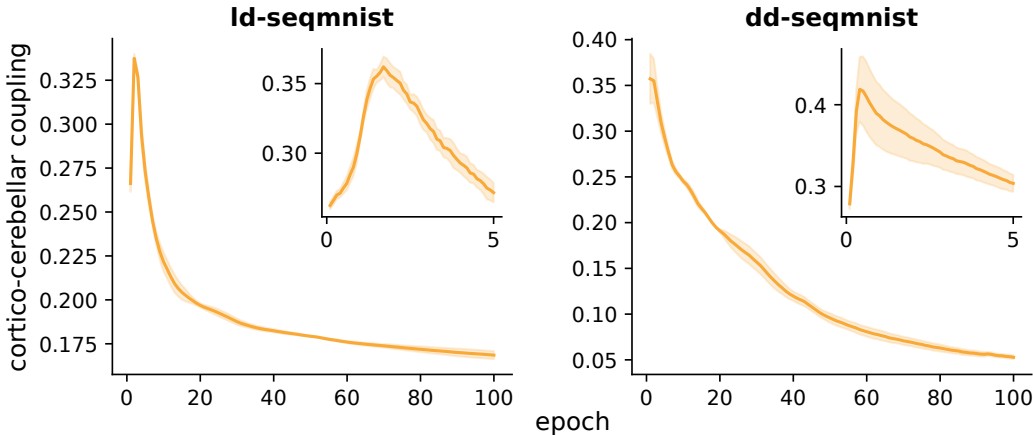

Figure S5: Evolution of cortico-cerebellar correlations (average absolute Pearson correlation between an LSTM unit and unit in the cerebellar hidden layer). Same as Fig. 3b but for ld-seqmnist and dd-seqmnist tasks.

Code for this task was based on (but altered from) the code provided at `https://github.com/yunjey/pytorch-tutorial/tree/master/tutorials/03-advanced/image_captioning`.

|       | BLEU   | Rouge-L | METEOR | CIDEr  | SPICE  |
|-------|--------|---------|--------|--------|--------|
| cRNN  | 0.2288 | 0.4790  | **0.2114** | 0.6928 | 0.1418 |
| ccRNN | **0.2294** | **0.4800** | 0.2112 | **0.6940** | **0.1422** |

Table S2: Mean metric scores identifying similarity between model generated and gold standard captions for test data. Metrics considered are BLEU (BLEU_4, i.e. for $n$-grams with $n \leq 4$), Rouge-L, METEOR, CIDEr, SPICE [76–80]

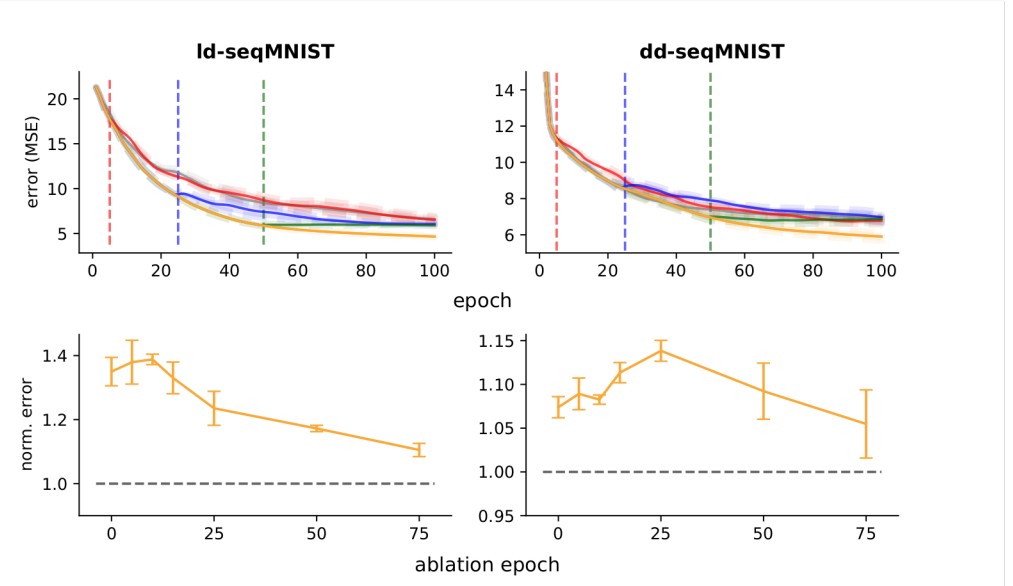

Figure S6: Effect of ablation for ld-seqMNIST (left) and dd-seqMNIST (right) tasks . Top row: learning curves for different ablation times. cRNN (i.e. ablation at epoch 0) in grey, ccRNN (no ablation) in orange, and other ablation times as given by the dotted vertical lines. Bottom row: model error normalised against (unablated) ccRNN for different ablation times.

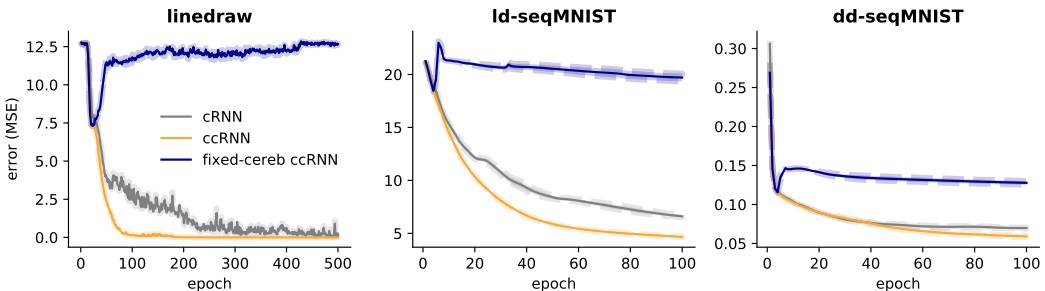

Figure S7: Learning with a fixed cerebellum. In this case cerebellar parameters are learnt for the first 10% epochs (to obtain non-zero weights) before being fixed for the remainder of training, forcing the RNN to use stale, out-of-date predicted gradients.

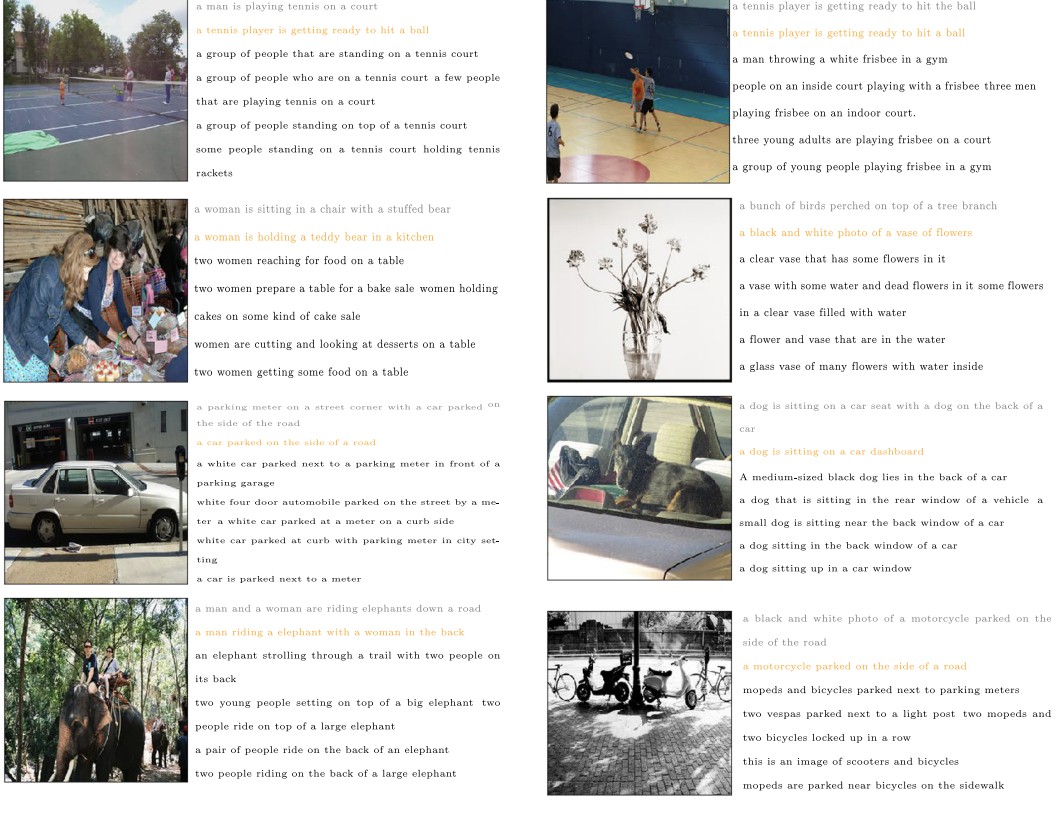

a man is playing tennis on a court

a tennis player is getting ready to hit a ball

a group of people that are standing on a tennis court

a group of people who are on a tennis court a few people that are playing tennis on a court

a group of people standing on top of a tennis court

some people standing on a tennis court holding tennis rackets

a tennis player is getting ready to hit the ball

a tennis player is getting ready to hit a ball

a man throwing a white frisbee in a gym

people on an inside court playing with a frisbee three men playing frisbee on an indoor court.

three young adults are playing frisbee on a court

a group of young people playing frisbee in a gym

a woman is sitting in a chair with a stuffed bear

a woman is holding a teddy bear in a kitchen

two women reaching for food on a table

two women prepare a table for a bake sale women holding cakes on some kind of cake sale

women are cutting and looking at desserts on a table

two women getting some food on a table

a bunch of birds perched on top of a tree branch

a black and white photo of a vase of flowers

a clear vase that has some flowers in it

a vase with some water and dead flowers in it some flowers in a clear vase filled with water

a flower and vase that are in the water

a glass vase of many flowers with water inside

a parking meter on a street corner with a car parked on the side of the road

a car parked on the side of a road

a white car parked next to a parking meter in front of a parking garage

white four door automobile parked on the street by a meter a white car parked at a meter on a curb side

white car parked at curb with parking meter in city setting

a car is parked next to a meter

a dog is sitting on a car seat with a dog on the back of a car

a dog is sitting on a car dashboard

A medium-sized black dog lies in the back of a car

a dog that is sitting in the rear window of a vehicle a small dog is sitting near the back window of a car

a dog sitting in the back window of a car

a dog sitting up in a car window

a man and a woman are riding elephants down a road

a man riding a elephant with a woman in the back

an elephant strolling through a trail with two people on its back

two young people setting on top of a big elephant two people ride on top of a large elephant

a pair of people ride on the back of an elephant

two people riding on the back of a large elephant

a black and white photo of a motorcycle parked on the side of the road

a motorcycle parked on the side of a road

mopeds and bicycles parked next to parking meters

two vespas parked next to a light post two mopeds and two bicycles locked up in a row

this is an image of scooters and bicycles

mopeds are parked near bicycles on the sidewalk

Figure S8: Example images from the validation set with corresponding model captions (cRNN in grey and ccRNN in orange) and gold standard captions (black). Here we show a combination of examples of how the models describe the presented image. In some case all or some models fail to give an accurate description of the image. In other cases all models are able to an accurate caption describing the image, with each model displaying subtle differences in the generated captions.