# OpenReview forum: "Cortico-cerebellar networks as decoupling neural interfaces"
_NeurIPS.cc/2021/Conference — NeurIPS 2021 Poster_

### Official Review · Reviewer_WvvH · 2021-07-14

**Rating:** 8
**Confidence:** 4

**Summary:**

During a network update step all layers need to wait for the feedforward pass (or its temporal RNN analogue) to reach "the end" and only then the error is calculated, gradient computed, weights updates. So effectively most of the time the lower layers are just waiting for the feedback to trickle down. This is called "feedback locking" and is inefficient. The paper proposes to add a second network to the architecture that learn to predict intermediate layer gradient from their activations, allowing an immediate update. The overall architecture bears hypothesized similarity to the role of cerebellum in primate brain.

I find the idea interesting and the execution faithful and detailed. My main concern is whether the chosen control experiment is doing a good job at eliminating confounding factors potentially responsible for the better performance of the proposed method (see below for detailed explanations and a proposal).

I am reserving my final assessment till after authors' rebuttal.

---- UPDATE ----
Authors' rebuttal has fully addressed my concerns and resolved them in a positive manner. I am now much more confident in the value of the contribution. Raising the score to "8: Top 50% of accepted NeurIPS papers, clear accept"

**Limitations And Societal Impact:**

The authors do provide some disclaimer statements regarding the limitations of their model and its predictions. It would be curious however to also learn authors' opinion about the limitations in (a) model's actual correspondence with the biological counterparts, (b) "failure" modes where the this model would not be applicable and might even be detrimental (if such cases can be identified). Is it always a good idea to approximate the gradient and provide this information mid-update or could it lead to unintended consequences in both biological and artificial netoworks?

**Main Review:**

General: What is the supporting evidence for the brain not being afflicted by the "feedback locking" problem. Do we know for certain that there is such a mechanism in place in the brain? Or could it be the case that the brain offers no solution for the "feedback locking" problem and this locking is actually part of normal operation?

88-89: Related to the question above. It would be ideal to see a reference to a in-vivo study where the up-stream portion of a certain brain "circuit" is being disabled so that we are certain the the signal is not reaching the higher level where the "loss" is computed (so that the credit assignment cannot happen). And then we feed some input to the lower portion of the "circuit" and confirm that is still learns.

General: If the cerebellum is believed to act as a producer of such a "synthetic gradient", doesn't this require the cerebellum to have direct connections to the part of the cortex where the "synthetic gradient" is (presumably) being used, say the motor cortex? But there is quite some distance, anatomically, between the motor cortex and cerebellum. Do such connections exist and if not, how it can still be responsible for carrying out the ascribed function?

General: There seems to be an issue of infinite regress - which brain structure will play the role of cerebellum for the cerebellum itself? Same applies to ccRNN - how do we avoid "feedback locking" when training the C network? I would imagine both the C and the Main network are being trained simultaneously so we cannot pre-train C in advance, can we?

201: "network needs to produce a straight line towards" - is the output of the network at each time t an (x, y) tuple of coordinates for the next target on the line? Perhaps this could be explicitly stated here for clarity.

209: Can we be certain that the positive effect of ccRNN here comes from the ability to avoid "feedback locking" and not because the C network is acting as a learning stabilizer / variance reducer, which then leads to more efficient learning? Here is a control experiment to consider: what if cRNN would also have a "cerebellum", but this time it would provide the learning signal only at the last time step? So instead of giving it directly to the RNN, we would pass it though the cerebellum network (but only at the last stage, preserving the "feedback locking" problem). If in this setup the resulting trajectories would still be more ataxious than ccRNN, then we would know that it is the precisely the ability to provide gradients and the intermediate steps is what is behind the improved performance, and not such factors as (a) the potential stabilizing effects of the C network, (b) the added overall system capacity (more weights), etc. What you imagine the outcome of such a control experiment would be? Could you run it in time for the rebuttal?

226: "pairwise" refer here to the pairs of what? Nodes? So the correlation between the activity of each node in the "cortical" network with each node of the "cerebellum"? At each time step?

252: Figure 4c: Error bars seem to be very small (compared to the ones on Figure 2c for examples). Given here the experiments was repeated only 3 times (compared to 10 before), one would expect longer error bars. Or was the results very consistent in this case? Why only 3 runs by the way, and not 10 as before?

**Time Spent Reviewing:**

3

---

> ### Author Response · Authors · 2021-08-09
> **Reply to reviewer WvvH**
>
> We thank the reviewer for the constructive and positive feedback. Below we provide a detailed reply to all the points raised.
>
> 1. *“What is the supporting evidence for the brain not being afflicted by the "feedback locking" problem”*
>
> **Reply**: We believe that our paper may be the first one to suggest that the brain indeed must address this problem which may happen through cortico-cerebellar interactions. This in turn can explain a number of experimental observations as demonstrated in our paper (e.g. faster learning, reduced ataxia behaviours and feedback interval dependency). So in light of our proposal these experimental observations can now be used as evidence for the locking problem in the brain. Moreover, it should be highlighted that this is a theoretical problem which exists for a large range of networks which perform gradient-driven credit assignment with spatio-temporal dependencies, as is the case in deep learning and potentially the brain (Richards et al. Nature Neuroscience 2019). Finally, it should be said that up to recently there was also virtually no evidence for the brain to perform backprop-like credit assignment, but new theoretical work combined with experimental evidence now suggests that the brain may approximate backprop (e.g. Payeur et al. Nature Neuroscience 2021, Sacramento et al. NeurIPS 2018).
>
> 2. *“88-89: Related to the question above. It would be ideal to see a reference to in vivo study where the up-stream portion is disabled…then feed input to the lower portion and confirm that it still learns.”*
>
> **Reply**: This is indeed one of the key experiments that would provide direct support to our study. But to the best of our knowledge, this has not been done. In part because it is only recently that this is feasible with the advent of optogenetics and the recent interest in cortico-cerebellar interactions. We hope that our study will be used to guide such experimental work. Indeed together with collaborators we are currently aiming to test some of these predictions made by our model. We note, however, that this is non-trivial, and based on the model this would only happen if the cerebellum has learnt a good enough approximation, which would be hard to show experimentally in the same set of experiments knocking-out a particular brain circuit. We will add this as a prediction to be tested experimentally.
>
> 3. *“If the cerebellum is believed to act as a producer of such a "synthetic gradient", doesn't this require the cerebellum to have direct connections to the part of the cortex where the "synthetic gradient" is (presumably) being used, say the motor cortex? But there is quite some distance, anatomically, between the motor cortex and cerebellum. Do such connections exist and if not, how it can still be responsible for carrying out the ascribed function?”*
>
> **Reply**: Thank you for highlighting this. In section 2.1.2 we describe how our systems level model relates to a more biological plausible consideration. More specifically, we discuss how the cerebellum provides feedback connections via higher order thalamocortical loops that target the apical dendrites of pyramidal cells in the neocortex (lines 165-166). On the one hand this is consistent with our model since these feedback loops provide a good substrate for driving cortical learning through cortical feedback prediction. On the other hand this is consistent with computational models of biological plausible feedback, which use distal dendrites to encode backprop-like error signals ([Payeur et al. Nature Neuroscience 2021](https://www.nature.com/articles/s41593-021-00857-x)
> , [Sacramento et al. NeurIPS 2018](https://papers.nips.cc/paper/2018/hash/1dc3a89d0d440ba31729b0ba74b93a33-Abstract.html)). Although we have not modelled explicitly these cortico>pons>cerebellar (and respective cerebellar>thalamic>cortical) connections, we did discuss this as one of the interesting future directions, which might themselves play an important role (information filtering) in the proposed cerebellar-mediated unlocking (lines 315-321).
>
> 4. *“There seems to be an issue of infinite regress..how do we avoid "feedback locking" when training the C network?”*
>
> **Reply**: This is in principle a problem, however the original DNI paper proposes an elegant solution to this by training the separate network (the cerebellum in our case) using its own predicted gradients -- this is known as bootstrapping in the ML literature.  Here we follow the same principle. For the temporal case (focus of our paper), the cerebellar target gradient delta-bar is constructed via a combination of cortical-derived-gradients and ‘bootstrapped’ cerebellar self-predictions (C(h_T) in line 104). The latter enables the cerebellum to consider all future loss gradients whilst its feedback lock is only constrained to the current BPTT truncation. An interesting and relevant comparison can be found in reinforcement learning algorithms which employ bootstrapping to estimate a value function (e.g. temporal difference or Q-learning). For the spatial case we envisage a similar possibility by attaching distinct cerebellar networks to different neocortical areas, so that the cerebellum only has to wait for later (in space) cerebellar signals for learning and does not have to wait for neocortical processing to finish (c.f. ‘full unlock’ described in SM A). These ideas are related to recent experimental evidence showing that different parts of the cerebellum encode different tasks ([King et al. NatureNeuro 2019](​​https://www.nature.com/articles/s41593-019-0436-x)), the modular structure of the cerebellum ([Apps et al. Cerebellum 2018](https://dx.doi.org/10.1007%2Fs12311-018-0952-3)) and time-dependent plasticity rules at the Granule cell/Purkinje cell synapse ([Suvrathan et al. Neuron 2016](https://doi.org/10.1016/j.neuron.2016.10.022)). We will add these points to the discussion.
>
> 5. *“I would imagine both the C and the Main network are being trained simultaneously so we cannot pre-train C in advance, can we?”*
>
> **Reply**: This is an interesting suggestion. It might indeed be possible to pre-train C to predict a range of gradients, but we think that this setup would be hard to generalise well enough to a real task. However, a scenario in which we combine pre-training with training during the actual task might be promising, and could be explored in future work.
>
> 6. *“201: ‘network needs to produce a straight line towards’ - is the output of the network at each time t an (x, y) tuple of coordinates for the next target on the line?”*
>
> **Reply**: Yes that is correct. A 2D line where the target at time t is the tuple $(x_t, y_t)$ -- i.e. the desired point at a given point in time. We agree this could be stated more clearly and will amend.
>
> 7. *“209: Can we be certain that the positive effect of ccRNN here comes from the ability to avoid "feedback locking" and not because the C network is acting as a learning stabilizer / variance reducer...what would you imagine the outcome of such a control experiment would be?”*
>
> **Reply**: Thank you for these suggestions. A priori, since the cerebellum is innately forward looking, we believe its role is most important during intermediate timesteps where the main network would not by itself be able to consider future loss gradients. Restricting cerebellar output to only the last timestep should therefore have a severe impact on the original improvements seen in the ccRNN. To verify this, we have run two extra control models for all but the last task (due to time constraints). The first control is a ccRNN model in which the C network only provides gradients at the final timestep, as you suggest (end-only ccRNN). The second control aims to highlight the importance of accurate predicted gradients (c.f. learned alignment to true gradients in Fig S6); here we apply predicted gradients in the same way as ccRNN (each truncation window) but the weights of the cerebellar network are now fixed once 5% of training time is completed (epoch 25, 5 for linedraw/seqmnist tasks respectively), leading to stale, ‘out of sync’ gradients (fixed-cereb ccRNN). You can find these new results here: https://imgur.com/a/cp6azsl. We observe both cases to have a significant negative impact on the original ccRNN model.
>
> 8. *“226: ‘pairwise’ refer here to the pairs of what...activity (of respective networks) at each timestep?”*
>
> **Reply**: It refers to pairs of nodes from the cortical RNN (LSTM cell/output state cells) and (hidden) nodes from the cerebellar network. Population activity is recorded at the end of each truncation window since it is at these points that cerebellar activity is being used by the RNN. This is explained in more detail in the SM (lines 663-667). We agree this could be expressed more clearly in the main text and will amend it.
>
> 9. *“252: Figure 4c: Error bars seem to be very small...why only 3 runs by the way, and not 10 as before?”*
>
> **Reply**: We think that the relatively small error bars in Fig 4c compared to 2c are partly due to the significantly greater number of updates in the seqMNIST tasks compared to the line drawing task (960 batches/epoch vs 100 batches/epoch respectively). With more updates might come less dependence on initial conditions and hence smaller error bars across seeds, though we agree they are perhaps surprisingly small. With regards to why we use 3 seeds for the seqMNIST tasks compared to 10 seeds for line drawing, it was simply due to time constraints. There were quite a few different hyperparameter configurations and each run required significantly more compute time compared to the more simple line drawing task. We have now tested the main seqMNIST conditions (Fig 4) over 10 seeds and observe the same results. We will include these new runs in the new version of the manuscript.

---

> > ### Author Response · Authors · 2021-08-09
> > **Reply to reviewer WvvH (part 2)**
> >
> > Due to character limit we address the final point (re Limitations And Societal Impact section) in this separate comment.
> >
> > 10. *"The authors do provide some disclaimer statements regarding the limitations of their model and its predictions...would be curious to also learn authors' opinion about the limitations in (a) model's actual correspondence with the biological counterparts, (b) ‘failure’ modes where the this model would not be applicable and might even be detrimental."*
> >
> > **Reply**: We will use the extra space to include more limitations. Regarding (a):
> >
> > A.1 As discussed above, we do not model the long-range projections via the thalamus and pons that mediate cortico-cerebellar interactions. Although it is possible that this might lead to a more efficient decoupling, we have not tested it and it is currently an important limitation. This is discussed in lines 315-321.
> >
> > A.2 In our modelling we assume that gradients can be provided by the cortex in some form to the cerebellum. In addition, we also assume that the cerebellum can influence learning directly through approximated gradients. The second assumption has some support as it is known that cerebellar-thalamic-cortical projections can target distal dendrites, which have recently been proposed to encode gradient/error signals ([Payeur et al. Nature Neuroscience 2021](https://www.nature.com/articles/s41593-021-00857-x)
> > , [Sacramento et al. NeurIPS 2018](https://papers.nips.cc/paper/2018/hash/1dc3a89d0d440ba31729b0ba74b93a33-Abstract.html)). We discuss this in lines 161-166.
> >
> > Regarding (b):
> >
> > B.1 Although the same framework can be used to model an inverse view of cerebellar function (see appendix A in SM for discussion) our results have focused solely on the forward cerebellum model (feedback unlocking), so it is not clear exactly what observations we could reproduce with that view of the cerebellum. The clear difference between these two modes is that one drives learning directly, while the other speeds up inference directly. As suggested by the reviewer above, a key experiment to perform to tease these two apart is to inhibit feedback/feedforward processing from specific brain regions to suppress the feedback or feedforward unlocking of the cerebellum.
> >
> > B.2 One of the new controls that we performed in which the cerebellum does not learn (see fixed-cereb ccRNN here: https://imgur.com/a/cp6azsl) demonstrates that if learning in the cerebellum cannot keep up (e.g. by impairing the inferior olive) with the main network the gradients produced by the cerebellum can have a strong negative impact in learning. There is recent experimental evidence to support this ([Welsh & Harvey 2001 EJN](https://doi.org/10.1046/j.1460-9568.1998.00400.x) showing that impairing the inferior olive blocks learning). Other perturbations that would impair cerebellar learning would have an important impact on learning in the main network.  This suggests that the cerebellum must have evolved to adapt quickly, which may be consistent with its feedforward architecture, sparse connectivity and modular structure.

---

> > > ### Comment · Reviewer_WvvH · 2021-08-26
> > > **A perfect rebuttal! My concerns were addressed, raising the score.**
> > >
> > > I thank the authors for a very thoughtful, carefully crafted and engaging rebuttal!
> > >
> > > While reading the response and re-engaging myself in the the discussion I've realized that I actually would be very much interested in seeing this question being explored further to confirm/reject the hypothesis that a mechanism similar to the one being proposed is exists in a primate brain. Such an interest in my opinion speaks in favor of accepting this work to NeurIPS making it visible to the wider NS+ML community.
> > >
> > > re 209: Thank you very much for running the additional control experiments! In my mind the new results indeed confirm that the observed gain is attributable to "fixing" the feedback locking issue.

---

> > > > ### Author Response · Authors · 2021-08-26
> > > > **reply**
> > > >
> > > > We would also like to thank you for the constructive review interaction and reading our (long) rebuttal.
> > > >
> > > > The feedback locking problem in the brain is also something that we have learnt to appreciate throughout this project. Glad that you can also appreciate this as an interesting problem.

---

### Official Review · Reviewer_pGCk · 2021-07-15

**Rating:** 7
**Confidence:** 3

**Summary:**

The authors propose to interpret the function of the cerebellum through the recently developed theory of decoupled neural interfaces, referred to as DNI, allowing for online training of a deep neural network and not requiring locking. After introducing the DNI, the paper extensively explains the possible mapping between the model and observe cortico-cerebellar networks, leading to a clear explanation of how they conceive feedback to be implemented in the brain. Finally, the paper proposes to compare the results obtained with their architecture on a variety of tasks validating their model.

**Limitations And Societal Impact:**

Limitations and possible research directions are nicely described.

**Main Review:**

The paper is well-written and proposes a rather uncommon way of addressing feedback in the ML/DL literature. It is highly appreciated that the paper clearly delineates structural properties, i.e., mapping onto known neural structures and the model of computation and learning.

However, an important weakness of the paper is that it does not try to understand the nature of the connections and neurotransmitters that would allow for such a model to be viable. The mapping only vaguely covers these aspects and weakens the paper. It would have been appreciated that either the paper itself a "biologically plausible" model and structure rather than forcefully mapping it onto the cerebellum and cortical networks. That does certainly not invalidates the point of the paper, but even with mitigating those arguments, the paper would remain strong while avoiding some oversimplification.

One could have also appreciated comparison to other models than itself in the experiments. It is not possible to know how well or poorly it performs compared to existing non-biologically plausible alternatives either SOTA or simply relevant to the paper.

**Time Spent Reviewing:**

4

---

> ### Author Response · Authors · 2021-08-09
> **Reply to reviewer pGCk**
>
> We thank the reviewer for the constructive and positive feedback. Below we provide a detailed reply to all the points raised.
>
> 1. *“an important weakness of the paper is that it does not try to understand the nature of connections and neurotransmitters for the model to be viable...it would appreciated that either the paper itself a "biologically plausible" model and structure rather than forcefully mapping it onto the cerebellum and cortical networks.”*
>
> **Reply**: We agree that it would be nice to have a more biologically plausible model, which models the different elements in a less abstract way. However we have decided on this first paper to take a systems perspective on the problem of cortico-cerebellar interactions which allows us to have a more functional/higher level model which can already capture long-standing observations in the field. This also means that we can build a first mapping between neuroscience and existing deep learning methods, opening up a field of research, in which future studies can explore many exciting questions such as the one raised by the reviewer. With regard to the nature of the connections, in section 2.1.2 we discuss how the cerebellum provides feedback connections via higher order thalamocortical loops that target the apical dendrites of pyramidal cells in the neocortex (lines 165-166). On the one hand this is consistent with our model since these feedback loops provide a good substrate for driving cortical learning through cortical feedback prediction. On the other hand this is consistent with computational models of biological plausible feedback ([Sacramento et al. NeurIPS 2018](https://papers.nips.cc/paper/2018/hash/1dc3a89d0d440ba31729b0ba74b93a33-Abstract.html); [Payeur et al. Nature Neuroscience 2021](https://www.nature.com/articles/s41593-021-00857-x)). With regard to the nature of neurotransmitters (e.g. excitatory/inhibitory neurotransmission) we agree this would be an interesting extension to our model but we think it is beyond the scope of this first paper.
>
> 2. *“One could have also appreciated comparison to other models than itself in the experiments…either SOTA or simply relevant to the paper.”*
>
> **Reply**: To the best of our knowledge there are no other models that could be easily mapped onto the cerebellar-cortical architecture. In addition DNI (from which the model is built) is already considered to be one of the SOTA methods for learning in RNNs when in the presence of  short/reduced BPTT windows. Explicit comparisons to other ML algorithms have already been achieved (see original [Jadeberg et al. PMLR 2017](http://proceedings.mlr.press/v70/jaderberg17a.html) or [Marschall et al. JMLR 2019](https://par.nsf.gov/biblio/10219212)) and we felt the inclusion of such results would not be novel or relevant enough for the main story of the paper. There are more recent algorithms designed to perform unlocking (e.g. [Lee et Lee IJCNN 2019](https://ieeexplore.ieee.org/abstract/document/8851709?casa_token=s-9OGZLmLYcAAAAA:_JHi93keBnOQUUwd8IY0rLntfL4gfH9Rs0dnhbdGnKl0UkykPXLzLXTOxBT0SOiVjKdsZ83Dpg); [Belilovsky et al. PMLR 2020](http://proceedings.mlr.press/v119/belilovsky20a.html); [Zhuang et al. IEEE 2021](https://ieeexplore.ieee.org/abstract/document/9399673)), but those are more complex algorithms which would require further biological assumptions and it is currently unclear how they might generalise to the temporal domain.

---

> > ### Comment · Reviewer_pGCk · 2021-08-17
> > **Response**
> >
> > Thank you for the response. I would appreciate it if the above discussion provided by the authors could be added to the paper.
> > I still believe this is a good paper and should be accepted.

---

> > > ### Author Response · Authors · 2021-08-17
> > > **Response to pGCK**
> > >
> > > Thank you for reading and replying to our response. We will include these points in the paper as much as possible.

---

### Official Review · Reviewer_YMU3 · 2021-07-15

**Rating:** 8
**Confidence:** 4

**Summary:**

In this work, the authors propose a neural architecture inspired by the anatomy of the cortico-cerebellar pathways in the brain.  The method relies on a previously proposed framework of decoupling neural interfaces. Here, the authors suggest that the cerebellar circuit calculates the future error of the cortical representation. It allows the implementation of Back Propagation Through Time without the phase-lock problem—where the network has to be wither in a feedforward of a feedback phase to calculate the local weight gradients.

**Limitations And Societal Impact:**

yes

**Main Review:**

**Strengths**
- Considering the cerebellum as a circuit that calculates the expected error is a promising avenue that aligns with many current ideas about the cerebellum function.
- The decoupling neural interfaces framework is a concrete model that could be used in the future to study cerebellar function.
- The authors show the benefits of the architecture on various tasks that would (potentially) be associated with the cerebellum.

**Concerns and weaknesses**
- The authors show that the methods work well (or at all) only when there are intermediate inputs of external feedback. It seems to me that the role of the cerebellar layer here is to make sure the output is interpolated smoothly between the supervised feedback. It would also explain the higher error in the non-smooth outputs.
- I found the details on the temporal feedback in RNNs (lines 96-108) hard to follow. In particular, the loss function for the temporal feedback. In here, delta-hat is the feedback from the cerebellar layer; why does the cerebellar activity also appear explicitly within the expression for delta-bar? This section should be revised and better explain the loss function being used; it is fundamental to this work and should not be left for a reference to make sense.
- The feedforward-backward phases of training are only one problem of several problems of implementing  BP and BPTT in a biological realistic network. While the paper does not mention other substantial issues, I don't

**Time Spent Reviewing:**

6

---

> ### Author Response · Authors · 2021-08-09
> **Reply to reviewer YMU3**
>
> We thank the reviewer for the constructive and positive feedback. Below we provide a detailed reply to all the points raised.
>
> 1. *“..methods work well only when there are intermediate inputs of external feedback.”*
>
> **Reply**: We agree that the addition of cerebellar provided synthetic gradients is most helpful when external feedback is only available intermittently; this is true both for the error during training (Figs 3a, S2) and also in the error for the interpolated points (i.e. ataxia score, equivalent summary plot not shown). We do however point out that the cerebellar network has a (albeit smaller) positive effect even for very dense supervised feedback (e.g. external feedback interval = 1, Fig. 3a).  We agree that these improvements arise from the main network having a more forward looking view which encourages "smoother" network output.
>
> 2. *“details on the temporal feedback in RNNs (lines 96-108) hard to follow.”*
>
> **Reply**: Thanks for raising this lack of clarity. The main point here is that the cerebellar network can avoid the potential issue of having to wait for all future timesteps/losses to occur before its target (delta-hat) is defined. This is achieved by incorporating only the nearby ‘true’ future error gradients (within the cortical feedback window) alongside an estimation of error gradient beyond that window; the latter itself is a bootstrapping term computed via the cerebellum (C(h_T) in line 104; also highlighted in Fig. 1b by small red arrow in the cerebellar loss function box). An interesting analogy is to that of the value function in reinforcement learning, where the target also depends on a future self-estimation term (i.e. $V(S_t) \leftarrow R_{t+1} + \gamma V(S_{t+1})$). The result is that the cerebellum can now update at each feedback window, and does not require backpropagation through time to take place beyond the window length.  We appreciate these lines (96-108) are fairly dense and will clarify the text.
>
> 3. *“The feedforward-backward phases of training are only one of several problems of implementing BP and BPTT in a biological realistic network...”*
>
> **Reply**: It seems that the reviewer did not finish writing down the point. Can you please complete your point? We appreciate that this is only one of the open issues in the field of mapping BP/BPTT to biology, and can highlight this point in the final discussion.

---

### Official Review · Reviewer_cVWr · 2021-07-16

**Rating:** 8
**Confidence:** 3

**Summary:**

This paper builds on past work that uses synthetic gradients (estimated by small networks) to decouple parts of a network so that they can run in parallel. The current paper proposes that the cerebellum may produce such synthetic gradients to deal with delayed feedback to recurrent networks. To illustrate how this could work, the authors model a cortico-cerebellar network as a LSTM (cortex) and feedforward gradient-estimating network (cerebellum). They show that the system can learn various tasks, including line and digit drawing and caption generation. The further show that disrupting the cerebellar input leads to something resembling ataxic behaviour due to cerebellar damage.

**Ethical Concerns:**

The work does not raise ethical concerns.

**Limitations And Societal Impact:**

The work has a minor societal impact. In the checklist, the authors say they have discussed limitations in sections 2.1.2 and 4, but I don’t see such a discussion.

**Main Review:**

Originality:
As far as I know, this is a new idea. It combines the well-established neuroscience concept of the cerebellum as a shallow feedforward supervised network and a recent approach in deep learning.

Quality:
The work seems to be technically sound and suitably thorough.

I have a reservation related to the analogy between the synthetic gradient network and the cerebellum. Reversible cerebellar lesions not only affect learning, but induce ataxia in previously learned movements (e.g. Hore, J., & Flament, D. (1988). Changes in motor cortex neural discharge associated with the development of cerebellar limb ataxia. Journal of neurophysiology, 60(4), 1285-1302). This seems to conflict with Figure 3c, which shows that the model isn’t sensitive to cerebellar ablation after a certain point in learning.

Clarity:
The writing is very clear and the paper is well organized. Some methodological details could be moved to the main text, e.g. the network architecture and the number of time steps of the line-drawing movements. I don’t think the plant that carries out the movements was ever described. Is it a 2D point mass with force inputs?

Significance:
As the authors point out, if this idea turns out to be physiologically realistic, it will provide an important new perspective on the cerebellum. It could also point to the physiological details of the cerebellum as a source of inspiration for future machine learning work on decoupled neural interfaces. The paper gives some preliminary evidence of physiological relevance of the model, but much more would be needed to make a convincing case (more than would be expected in a single paper). So I think the significance is unclear at this point, but it is promising.

Minor:
I believe the cerebellum provides input mainly to frontal, cingulate, and a few parietal areas (Strick, P. L., Dum, R. P., & Fiez, J. A. (2009). Cerebellum and nonmotor function. Annual review of neuroscience, 32, 413-434.) Could the authors comment on whether this restriction relates to the proposed role of cerebellum?

In Figure 2d, my understanding is that the “cortical temporal feedback %” applies to both cRNN and ccRNN. So cRNN works better if there is a long enough feedback window, otherwise synthetic gradients are helpful. If this is right, I’d suggest expanding the caption slightly as I missed it the first time.

Line 126: “our model suggests that the cerebellum should encode error signals, consistent with neural recordings showing cerebellar-encoding of kinematic errors during motor learning” I think the model suggests that the cerebellum should encode error gradients, which may only be consistent given additional assumptions.

Line 226: “pairwise correlations between all the neural activity in the main cortical network and the neural activity of the cerebellar module (see SM).” I had to check the SM to understand this, but a slight rephrasing would make it clear in the main text.


**Time Spent Reviewing:**

4 hours

---

> ### Author Response · Authors · 2021-08-09
> **Reply to reviewer cVWr**
>
> We thank the reviewer for the constructive and positive feedback. Below we provide a detailed reply to all the points raised.
>
> 1. *“reservation related to the analogy between the synthetic gradient network and the cerebellum...cerebellar lesions induce ataxia in previously learned movements...this seems to conflict with Figure 3c, which shows that the model isn’t sensitive to cerebellar ablation after a certain point in learning.”*
>
> **Reply**: Thanks for pointing this apparent inconsistency out. We should highlight that this is only true for a task which can be fully learnt as is the case of the simplified line drawing task (Fig. 2). For a more realistic task, as is the case of the seq-MNIST tasks, then even ablating at the end of the task can damage learning and can lead to ataxia-like behaviours. We believe that this could be because the task is never fully learnt (i.e. error > 0) and the parameters of the main network are therefore still pursuing, via optimisation (i.e. with non-zero gradients), some trajectory in parameter space which continues to depend on cerebellar provided predicted gradients. Our ablation results in the seq-MNIST tasks that we provided in Fig. S4 (Supp. material) demonstrate this. You can also find a comparison of the ablation results in the following summary plot (including results for standard seq-mnist, which were not included in the original paper): https://imgur.com/a/5XliyY3.
>
> Another possible interpretation here is that the paper cited by the reviewer highlights the role of the cerebellum in constructing inverse models, in which the cerebellum is involved in the generation of motor commands directly that are sent to downstream targets. In our model we focus on the forward model view of the cerebellum (mapped onto backward unlocking), but the same framework can be used to  model the inverse view of cerebellar function (see appendix A  describing this case and its mapping onto the cerebellum). Both inverse and forward views of cerebellar function are important for motor control and motor learning, although they could play a different role. Our model backs up the evidence that forward models could play a more important role when adapting to different dynamics while inverse models play more of a role in execution. Here we decided to focus on the forward model due to the growing evidence of its strong implications in more non-motor functions of the cerebellum.
>
> 2. *“methodological details could be moved to the main text, e.g. the network architecture and the number of time steps of the line-drawing movements...is it a 2D point mass with force inputs?”*
>
> **Reply**: Thanks for the suggestions. We will use the additional page to include some of these experimental details in the main text. Regarding movement plan, in our model (simple line drawing task) the planning is done directly by the RNN together with a read-out that maps the RNN output onto a 2D coordinate given a sensory cue (input) at the first timestep (Fig. 2a and main text). We will clarify this further in the main text.
>
> 3. *“The paper gives some preliminary evidence of physiological relevance of the model, but much more would be needed to make a convincing case.”*
>
> **Reply**: We agree, indeed we are currently exploring new DNI variants directly inspired by cerebellar features. We will briefly discuss this new line of research in the main text.
>
> 4. *“I believe the cerebellum provides input mainly to frontal, cingulate, and a few parietal areas…(does) this restriction relate to the proposed role of cerebellum?”*
>
> **Reply**: The restrictions in anatomical connections provided by Strick et al. was obtained in monkeys. However, there is evidence from fMRI studies in humans that the cerebellum is functionally connected to virtually all brain areas  including the temporal cortex supporting phonological processing, which is more pronounced in human primates. Cerebellar inputs into neocortical areas thus vary across species indicating they are shaped by evolution. For example, in human and non-human primates the cerebellum is known to connect to the prefrontal cortex, however this is still heavily debated in rodents. Taken together this suggests that the cerebellum provides an universal role across all cortical areas, which is consistent with our model, which can be applied to facilitate computations in virtually any brain area. Indeed, in our paper we provide a few examples that would be applicable to different brain areas (e.g. motorvisual computations and language understanding). Having said this, the proposed role would be particularly important for brain areas that must solve difficult temporal credit assignment tasks. We will clarify this point in the main text.
>
> 5. *“In Figure 2d, my understanding is that the “cortical temporal feedback %” applies to both cRNN and ccRNN..suggest expanding the caption slightly.”*
>
> **Reply**: Yes that is correct, we will clarify this.
>
> 6. *“126: cerebellum should encode error gradients, which may only be consistent (‘to cerebellar-encoding of kinematic errors’) given additional assumptions.“*
>
> **Reply**: Thanks for pointing this out. We agree and will clarify the link between errors and error gradients.
>
> 7. “226. ‘pairwise correlations’...I had to check the SM to understand this, but a slight rephrasing would make it clear in the main text.”
>
> **Reply**: Thanks for pointing this out. We agree this could be better worded. We will expand on the current description.
>
> 8. “authors say they have discussed limitations in sections 2.1.2 and 4, but I don’t see such a discussion.”
>
> **Reply**: We do highlight a limitation in section 2.1.2 regarding the generation of gradient signals for which we use BPTT, which is biologically implausible (​​”Our systems level model uses BPTT to generate temporal feedback signals.”) and discuss how this could be addressed. In section 4 we highlight two more limitations: (i)  “in the present study we do not explicitly model the brain structures that exist in between cerebellum and the neocortex (pontine nuclei and thalamus)” and (ii) “here we have focused on a variant of the model that predicts feedback signals needed for learning, but it is likely that the cerebellum is also involved in speeding up feedforward processing as in forward DNIs (see SM)". We will clarify these limitations.

---

> > ### Comment · Reviewer_cVWr · 2021-08-25
> > **Great paper**
> >
> > Thanks for the clarifications and improvements. In light of these and also of the responses to other reviewers I will increase my score.

---

### Decision · Program_Chairs · 2021-09-27

**Decision:**

Accept (Poster)

**Comment:**

This paper proposes that the previously published synthetic gradient approach, decoupled neural interfaces (Jaderberg et al. 2017), may serve as a model of the cortico-cerebellar learning. The reviewers found the proposed connection between DNI and cortico-cerebellar learning to be a promising lead for future biological investigation.  Since understanding cerebellar function and role in human cognition is a deep and important problem, the initial proposal of this possible connection to a recent AI approach is considered a valuable contribution.  While this connection is not validated in this work, the connection is considered in light of what is already generally known about the cerebellum, and the work explores the consequences of this comparison.

The results of the paper involve comparisons between a baseline RNN model which uses truncated BPTT and a ccRNN model that learns using the DNI mechanism.  While the DNI mechanism is not new, the tasks chosen are ones that are supposed to provide further illustrate the implications of the connection to cortico-cerebellar learning.

Personally, I found the submission to be creative in identifying the potential relationship between the existing ML model and the architecture of the brain.  I would emphasize that the ML portion is not new, so the contribution of this paper is specifically the proposal of how the DNI approach might serve as an analogy to cortico-cerebellar learning.  I initially found some details, especially around the presentation of the results, a bit unclear.  However, the authors and reviewers engaged during the discussion phase in a way that I believe will make the final version of the paper sufficiently clear.  Multiple reviewers raised their scores during the exchange.  I'm willing to endorse the reviewer consensus that this paper be accepted.